# Linking aerobic scope to fitness in the wild reveals potential opportunities to help recover imperiled salmon populations
Benjamin P. Burford [1,2] ✉, Brendan M. Lehman[1,2], Kenneth W. Zillig [3], Vanessa K. Lo[3], Alexandra G. McInturf[4], Garfield T. Kwan[3], Dennis E. Cocherell[3], Nann A. Fangue [3] & Cyril J. Michel [1,2]

Aquatic ectotherms are hypothesized to be vulnerable to warming and deoxygenation associated with environmental change because temperature and oxygen ($O_2$) supply can restrict aerobic scope (AS) in captivity. However, evidence of a direct association between AS and fitness in the wild is lacking, inspiring debate about the circumstances under which AS is the primary driver of population fluctuations. Using respirometry data, telemetry studies, long-term population monitoring, and in situ predator-prey experiments, we related AS to two Chinook salmon (*Oncorhynchus tshawytscha*) population bottlenecks in the wild, juvenile rearing and migration. We found that AS, which we quantified using the metabolic index ($\phi$), was associated with success probability for these bottlenecks only under a relatively narrow window of viable environmental conditions, depending on intraspecific metabolic trait diversity and hydrologic conditions. Opportunities for potentially high-impact temperature- and $O_2$-specific conservation and management actions using existing hydraulic engineering infrastructure could therefore exist when AS is between critical ($\phi_{crit}$) and stable ($\phi_{stable}$) values. Outside of this ecological threshold, changes in AS did not yield appreciable fitness benefits because successful rearing and migration were either exceptionally improbable (i.e., AS<$\phi_{crit}$), or seemingly independent of AS (i.e., AS>$\phi_{stable}$). In addition, AS impairments likely increased susceptibility to predation, and this may have been involved in the putative association between AS and fitness in the wild.

Aerobic scope (AS) is an organism's fundamental capacity to perform aerobic work above maintenance levels, given metabolic traits and environmental temperature and oxygen ($O_2$) supply[1,2]. For nearly a century, empirical studies have consistently shown that temperature and $O_2$ supply can restrict AS in captivity (e.g., refs. [1–3]). However, it has remained challenging to assess the extent to which AS restrictions govern ecological fitness[3], or the ability of individuals to grow, survive, and reproduce in the wild[4].

Restrictions to AS, as quantified by the metabolic index ($\phi$)—the temperature-dependent ratio of environmental $O_2$ supply and organismal metabolic $O_2$ demand[5]—have recently been implicated as a primary mechanism underlying past extinctions[6], contemporary biogeography[5,7], periodic range expansions[8], and future extirpations and fisheries collapses[5,9] in Earth's oceans. The main supporting evidence is that the warmwater

limits to the current distributions of marine ectotherms are bounded by critical $\phi$ values ($\phi_{crit}$), such that ectotherms do not inhabit marine environments unless they allow for AS at least two to five times greater than that required for life support (i.e., $\phi_{crit}$ typically ranges from 2 to 5 depending on the species evaluated)[5,10]. This has led to the conclusion that further limitations to AS imposed by warming and deoxygenation associated with climate change could make susceptible marine environments unviable by restricting ecological fitness[5,10]. However, this conclusion has not been validated because all relevant studies to date have assumed that AS is an important determinant of ecological fitness, rather than testing this relationship explicitly[3,10].

If AS is to be confidently used to identify ecological thresholds, or circumstances where small changes in environmental conditions produce

[1]University of California, Santa Cruz, Institute of Marine Sciences' Fisheries Collaborative Program, 1156 High Street, Santa Cruz, CA, USA. [2]Fisheries Ecology Division, Southwest Fisheries Science Center, National Marine Fisheries Service, National Oceanic and Atmospheric Administration, 110 McAllister Way, Santa Cruz, CA, USA. [3]Department of Wildlife, Fish and Conservation Biology, University of California Davis, Davis, CA, USA. [4]Coastal Oregon Marine Experiment Station, Oregon State University, Newport, OR, USA. ✉e-mail: bburford@ucsc.edu

large ecological responses[11,12], then it is crucial to understand how AS is related to ecological fitness. Reference 13 recently posited that strong ecological selective pressures have caused species to evolve to maximize aerobic performance under prevailing environmental temperatures and $O_2$ supplies. If this is the case, then $\phi_{crit}$ values could be an artifact resulting from species' tracking of preferred conditions for aerobic performance[3], rather than barriers to their distributions[10]. If species track preferred conditions for aerobic performance[13] rather than being pushed by AS barriers[10], higher AS will be required to prevent extinctions and extirpations, or recover imperiled populations, than $\phi_{crit}$. More generally, if AS only comprises one of many aspects of organismal performance rather than governing the entire suite, then AS may only be circumstantially relevant for fitness in the wild[14]. Accordingly, global AS thresholds (e.g., $\phi_{crit}$) might not be appropriate because the impact of AS on fitness could vary with environmental and life history demands[15].

Our objective was to investigate the extent to which AS could identify meaningful ecological thresholds by exploring if and how $\phi$ was related to two key aspects of ecological fitness: the ability of individuals to grow and to survive in the wild[4]. Some field-based evidence supports the hypothesis that AS can be restrictive of fitness in the wild: for example, unusually high temperatures and low $O_2$ have been implicated in mortality during migrations with potential population-level consequences[16–18]. Field-based evidence also supports the hypothesis that species have evolved to maximize AS: for example, interpopulation diversity in AS has been associated with local environmental conditions and migratory strategy[19,20]. Finally, observations that spawning adults and embryos tend to have reduced thermal tolerances compared to other fish lifestages[21] suggest that AS could only be circumstantially relevant for fitness in natural settings.

We hypothesized that, if AS is relevant for fitness in the wild, then any facilitative association would exhibit an ecological threshold delineated by both limiting and saturating circumstances[10,13]. In other words, the conditions under which environmental temperature- and $O_2$-dependent gains in AS facilitated measurable fitness benefits would occur between two values: $\phi_{crit}$, below which growth or survival was exceptionally improbable, and $\phi_{stable}$, a higher AS value where further AS gains did not result in fitness benefits. Furthermore, any difference between AS exhibited in captivity and $\phi_{stable}$ in the wild would reveal the difference between the fundamental and realized niche, given that AS is a fundamental characteristic, and that fitness is unlikely to scale proportionally with AS due to other ecological factors[15,22].

Using $\phi$, we examined how AS, alongside other environmental factors, was related to the ability of federally threatened and endangered Chinook salmon (*Oncorhynchus tshawytscha*) populations to traverse known bottlenecks in California's Sacramento-San Joaquin watershed. Here, Chinook salmon are at the warmwater limit to their freshwater range[23], and have been reduced to a fraction of their historical distribution, abundance, and diversity due to an era of rapid anthropogenic habitat modification beginning during the 1850s[24,25]. Chinook salmon have an anadromous lifecycle: individuals are born and spend early life in freshwater, migrate downstream to the ocean to feed and grow, and migrate back upstream to natal freshwater habitats to reproduce[23]. As such, low-elevation river deltas constitute a natural chokepoint that all individuals are obligated to negotiate regardless of the environmental conditions encountered. Temperature and $O_2$ fluctuations in the Sacramento-San Joaquin Delta (hereafter "Delta") are thereby likely to impact the success of two fitness-relevant behaviors of Chinook salmon—(1) habitat use for growth and development (i.e., rearing) and (2) migration—key bottlenecks hampering the recovery of remaining populations of this imperiled taxon[22,24–26].

Past studies of $\phi$ have mainly associated the biogeography of marine ectotherms with $\phi$ determined from monthly climatological temperature and $O_2$ at 1° latitude and longitude resolution[3,10]—in comparison, our approach had two main advantages relevant to our objective. First, ecological fitness was naturally measured along a gradient of $\phi$ at much higher ecological and spatiotemporal resolution than is currently feasible in the ocean. Second, the study system is equipped with infrastructure that enables direct control over environmental conditions according to wildlife

requirements, which is impossible in the ocean. For example, thermal stress thresholds for the eggs of endangered Chinook salmon[27] currently guide seasonal releases of cold water from the largest storage reservoir in California[28]. Furthermore, multiple studies have shown how reservoir releases could be used within this watershed to regulate downstream temperatures (e.g., refs. 29,30).

To progress the debate on the ecological importance of AS[3,14,15], and thereby facilitate the interpretation of $\phi$[3,10], our approach leveraged massive physiological and ecological datasets through three general steps. First, we parameterized the metabolic index to produce juvenile Chinook salmon-specific $\phi$ using metabolic traits derived from prior respirometry experiments of captivity-raised individuals ($n = 640$)[20,31–33]. Second, we assessed if and how $\phi$ was related to ecological fitness, which we defined as the probability of successful rearing and migration, using telemetry studies ($n = 2240$ tagged fish)[34] and long-term population monitoring ($n = 5401$ population surveys)[35] synchronized with environmental conditions[36]. And third, we evaluated how susceptibility to predation may have contributed to observed fitness detriments to juvenile Chinook salmon through an AS lens using in situ predator–prey experiments ($n = 2277$ predator–prey assays)[37–40] and a respirometry experiment of major predator species ($n = 35$ individuals)[32].

## Study system

We focused on fry and smolt freshwater lifestages of Chinook salmon for this study. The fry lifestage is characterized by rearing, which occurs in the warmer, more productive freshwater habitats[23] such as the Delta. The subsequent transition to the migratory smolt lifestage, which must traverse low-elevation freshwater habitats such as the Delta, is characterized by biochemical changes involved in osmoregulation and metabolism[23,41]. These changes prepare individuals for a pelagic lifestyle in the California Current, an Eastern Boundary Upwelling Ecosystem characterized by relatively shallow layers of cold, hypoxic water[5,8]. Because conserving intraspecific diversity is critical for the resilience of populations[42], individuals examined in this study were comprised of the four genetically distinct populations present in the Sacramento-San Joaquin watershed (winter, spring, fall, and late fall), each of which exhibits unique life history phenology[43]. While populations are named for the season when adults return to freshwater from the ocean, the timing of fry rearing in, and smolt migration through, the Delta is also somewhat distinct. During fall and winter, fry of winter-, spring-, and fall-returning adults are usually present, as well as smolts of winter-returning adults. During spring and summer, fry of late fall-returning adults and smolts of spring-, fall-, and late fall-returning adults tend to be present[30,44].

## Results and discussion
### Metabolic traits were specific to lifestage and population

In order to make specific estimates of $\phi$ given environmental temperature and $O_2$, we parameterized the metabolic index[5] (Eq. 1) with the metabolic traits of juvenile Chinook salmon. $\phi$ is a unitless integer directly proportional to factorial AS (FAS), or AS relative to standard metabolic rate (SMR), SMR being the energetic cost of maintenance and very limited movement in fishes[45].

$$\phi\ (i.e.,\ FAS) = \frac{O_2\ supply}{O_2\ demand \sim T} = \frac{O_2}{O_2 crit \sim T} = \frac{O_2}{\left(A/e^{\frac{-E}{k_B T}}\right)} \quad (1)$$

In this ratio, environmental $O_2$ supply is the partial pressure of $O_2$ in water (kPa), and organismal $O_2$ demand is the critical $O_2$ partial pressure ($O_2 crit$, kPa), or the minimum $O_2$ required for SMR. For ectotherms, $O_2 crit$ changes with temperature ($T$, K) according to the temperature dependence of reaction rates (i.e., the Arrhenius equation), where $A$ is the pre-exponential factor, $-E$ is the activation energy, and $k_B$ is Boltzmann's constant (eV). In the context of $\phi$, $A$, and $-E$ are termed "metabolic traits:" $A$ is hypoxia tolerance, or the natural logarithm (ln) of $O_2 crit$ at a theoretical

**Fig. 1 | Hypoxia tolerance deteriorated with temperature in juvenile Chinook salmon.** Association between hypoxia tolerance and temperature for **A** fry and **B** smolts. The intercept and slope of the associations between hypoxia tolerance and inverse temperature supplied the metabolic traits $A$ and $-E$, respectively, for each lifestage (Table 1). These traits were then used to parameterize the metabolic index (Eq. 1) so that environmental temperature and $O_2$ could be converted into $\phi$ (i.e., FAS). For the units to cancel out in this equation, it is necessary for temperature to take inverse format in eV; however, for the convenience of the reader, the secondary x-axis (above) shows the corresponding temperature in °C. In (**A**, **B**) points show data, while the gray lines and ribbons respectively show the fit and 95% bootstrapped CIs of a LMER with a by-population random intercept. Black lines are the fits of an MR with population included as a fixed effect. A higher $O_2$crit reflects poorer hypoxia tolerance, i.e., more $O_2$ is necessary for SMR.

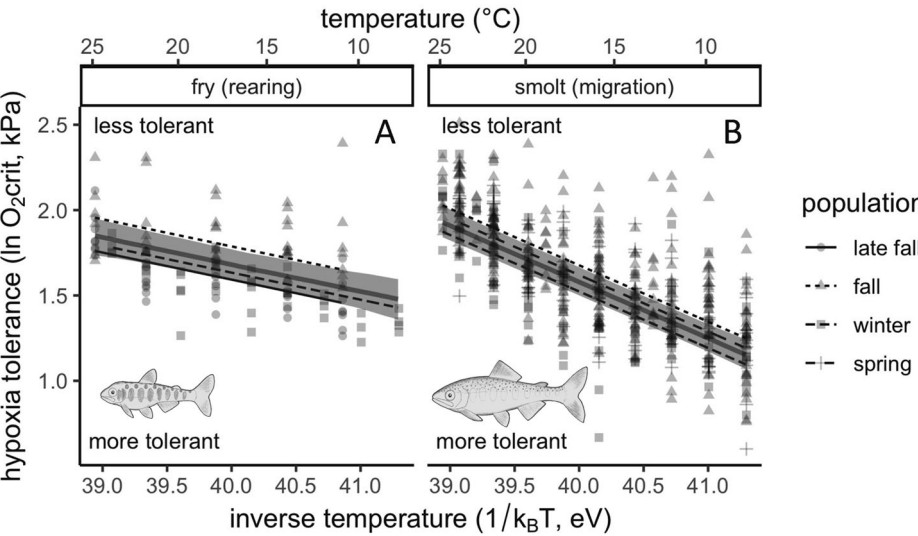

maximum temperature, and $-E$ is the temperature sensitivity of hypoxia tolerance[5,7]. Organisms with larger $A$ have greater hypoxia tolerance (i.e., less $O_2$ required for SMR), while organisms with more negative $-E$ have greater temperature sensitivity (i.e., lower activation energy of SMR).

Past efforts utilizing $\phi$ have assigned species-wide metabolic traits based on available data[3,10]. However, species are often comprised of portfolios of unique populations[42] and lifestages[23] and therefore may possess an assortment of distinct physiological sensitivities[19,21]. Consistent with other cold-water species[5,7], Chinook salmon fry and smolts of all examined populations showed a deterioration of $O_2$crit with increasing temperature—i.e., higher $O_2$ was required for SMR under warmer compared to cooler conditions (Fig. 1). Temperature sensitivities (±95% confidence intervals, CIs) for fry ($-E = 0.16 \pm 0.06$) and smolts averaged across populations ($-E = 0.33 \pm 0.03$) (linear mixed-effects regression [LMER]: conditional $R^2 = 0.54$, marginal $R^2 = 0.48$, residual df = 634) (Table 1 and Supplementary Equation 1) fell in the 20 and 50% quantiles, respectively, of a study of the thermal sensitivities of diverse ectothermic taxa[7]. This suggests that fish in our study had below-average or average values compared to other ectotherms. Within each lifestage, we found that populations slightly yet significantly differed in their hypoxia tolerance, but not in their temperature sensitivity (multiple regression [MR]: $R^2 = 0.52$, $F_{6,633} = 114.60$, $p < 0.001$) (Table 1 and Supplementary Equation 2). This could be due to differences in river conditions and migratory demands, which have been associated with intraspecific variation in AS among wild populations of adult sockeye salmon (*O. nerka*)[19], and potentially juvenile Chinook salmon[20,33]. Given the congruence of MR and LMER predictions (Fig. 1), we used metabolic traits averaged across populations (i.e., parameters from the LMER in Table 1) to produce $\phi$ for each lifestage that was broadly applicable across populations in the study system.

Previous research has reported a negligible or marginal impact of body size on hypoxia tolerance (e.g., refs. 5,8,46). In contrast, we found a substantial effect of body size on Chinook salmon $O_2$crit even over a relatively small size range (linear regression [LR]: $R^2 = 0.94$, $F_{1,638} = 9207$, $p < 0.001$) (Supplementary Fig. 1). Moreover, while controlling for the effect of body size, we found that smolts were more hypoxia tolerant (±95% CI) ($A = 14.8 \pm 1.07$ and $7.98 \pm 2.41$ for smolts and fry, respectively) and more temperature sensitive than fry (Fig. 1 and Table 1). The combination of high $A$ and low $-E$ equates to better hypoxia tolerance under cool conditions[7]. In other words, the $O_2$ demand of SMR for fry is relatively insensitive to temperature, allowing this lifestage to inhabit warmer or cooler waters with minimal impact to AS. The SMR of smolts, on the other hand, requires relatively higher $O_2$ when warmer (restricting AS), but relatively less $O_2$

when cooler (expanding AS). This ontogenetic variation likely reflects physiological adaptations to distinct environmental or energetic challenges[22]: fry tend to move from cooler upper river reaches to rear in warmer downstream reaches[30], while smolts seek cooler temperatures during downstream migration, the destination of which is a cold marine ecosystem with shallow layers of hypoxia[5,8,23]. Metabolic traits in our study were derived from captivity-raised fish; available evidence in fishes suggests that captivity acclimation can increase $O_2$crit[47], but to our knowledge it is unknown how the temperature sensitivity $O_2$crit changes with captivity acclimation. Nonetheless, our results reveal ontogenetic and interpopulation variation in metabolic traits for Chinook salmon, underscoring why conserving diversity for harvested species is critical for ecological and economic stability[25,42].

### Fitness benefits attributable to aerobic scope occurred within a relatively narrow subset of viable environmental conditions

Our results suggest that AS can both limit and facilitate the ecological fitness of Chinook salmon in the wild. Using $\phi$ specific to Chinook salmon fry and smolt lifestages averaged across populations (LMER parameters in Table 1), we assessed if and how AS, alongside other key environmental factors such as flow (volumetric discharge of the Sacramento River, the primary watershed feeding the Delta), was related to ecological fitness in the study system (Fig. 2A). Ecological fitness encompasses the ability of individuals to grow and survive[4]; we therefore examined both the probability that rearing wild fry occupied potential Delta habitat, and the probability that captivity-raised smolts survived through-Delta migration. Given that the former was not a direct measure of growth or survival, we assumed that the availability of suitable rearing habitat was related to the ability of fry to grow and survive, which has been shown in other regions of this watershed[48]. Both the utilization of rearing habitat (Fig. 2B) (generalized additive mixed-effects model [GAMM]: $R^2 = 0.24$, df = 23, residual df = 5379) (Supplementary Equation 3), and successful migration probability (Fig. 2C) (generalized additive model [GAM]: $R^2 = 0.18$, df = 17, residual df = 2423) (Supplementary Equation 4), showed inflections across the range of sampled $\phi$, revealing $\phi_{crit}$ and $\phi_{stable}$ for each life history bottleneck.

While $\phi_{crit}$ has been previously determined as the minimum monthly climatological $\phi$ within the contiguous range of a species[3,5], we determined $\phi_{crit}$ as the $\phi$ below which further declines in $\phi$ yielded no further measurable fitness detriment. Breakpoint analysis ($R^2 = 0.99$, relative standard error = 0.04 for both rearing and migration segmented regressions) revealed fry were unlikely to utilize a given region in the Delta for rearing if $\phi \leq 2.82 \pm 0.03$ (rearing $\phi_{crit}$, ±95% CI). Similarly, smolts were comparably

**Table 1 | Metabolic traits were specific to lifestage and population in juvenile Chinook salmon**

| Model | Parameter | Lifestage | Population | Value | Low 95% CL | Up 95% CL |
|---|---|---|---|---|---|---|
| LMER | $A$ | fry | NA | 7.98 | 5.57 | 10.35 |
| | | smolt | NA | 14.81 | 13.74 | 15.88 |
| | $-E$ | fry | NA | 0.16 | 0.10 | 0.22 |
| | | smolt | NA | 0.33 | 0.30 | 0.36 |
| MR | $A$ | fry | fall | 8.13 | 5.74 | 10.53 |
| | | | winter | 7.98 | 5.58 | 10.38 |
| | | | late fall | 7.94 | 5.55 | 10.33 |
| | | smolt | fall | 14.91 | 13.83 | 15.98 |
| | | | spring | 14.85 | 13.77 | 15.92 |
| | | | winter | 14.75 | 13.68 | 15.82 |
| | $-E$ | fry | fall | 0.16 | 0.10 | 0.22 |
| | | | late fall | 0.16 | 0.10 | 0.22 |
| | | | winter | 0.16 | 0.10 | 0.22 |
| | | smolt | fall | 0.33 | 0.30 | 0.36 |
| | | | spring | 0.33 | 0.30 | 0.36 |
| | | | winter | 0.33 | 0.30 | 0.36 |

Parameter values for $A$, (y-intercept), and $-E$ (slope) from two regressions: a LMER that compared $-E$ and $A$ between lifestage averaged across populations, and a MR that compared $A$ and $-E$ between lifestages and populations (Fig. 1). Data did not support the inclusion of $-E$ varying by population (Supplementary Table 1). Lower and upper 95% confidence limits (CLs) are shown ($p$ for all parameters <0.001).

unlikely to survive oceanward migration through the Delta if $\phi \leq 3.72 \pm 0.09$ (migration $\phi_{crit}$); however, it is likely that our estimate of migration $\phi_{crit}$ is conservative given the narrow range of sampled conditions in the smolt dataset.

In addition to $\phi_{crit}$, we found evidence of $\phi_{stable}$, which represents the environmental conditions beyond which further increases in AS did not guarantee an additional fitness benefit. Fitness probabilities increased with $\phi$ until a stable value of $3.81 \pm 0.04$ in fry (rearing $\phi_{stable}$) and $4.33 \pm 0.05$ in smolts (migration $\phi_{stable}$) was reached. Beyond $\phi_{stable}$, additional $\phi$ was not necessarily advantageous, and its influence on fitness likely waned compared to other environmental factors (e.g., flow). While stable values were only 35 and 16% higher than critical values for rearing and migration, respectively, and fell well within the range of $\phi_{crit}$ values reported for diverse species, some marine ectotherms that live extremely close to hypoxia tolerance limits show distribution shifts when environmental $O_2$ declines by $\leq 1\%$[49]. Thus, in addition to limiting distributions, our results suggest that small differences can also be relevant in how AS enhances fitness.

Flow is an important hydrologic variable for the behavior and ecology of Chinook salmon, particularly in the study system[24–26]. Using interactions in the rearing GAMM and migration GAM, we investigated how AS (a metric of water quality) and flow (a metric of water quantity) synergistically impacted fitness. While flow did not change $\phi_{crit}$ or $\phi_{stable}$ for either rearing or migration, it affected the baseline probability of fry rearing (Fig. 2D) and smolt migration success (Fig. 2E) through a compensatory relationship with $\phi$. Specifically, below $\phi_{crit}$, exceptionally high flows ($\geq 80\%$ quantile) propped up success probabilities; between $\phi_{crit}$ and $\phi_{stable}$, success probability under lower $\phi$ was maintained via higher flows; and above $\phi_{stable}$ relationships were mixed. Habitat quality and quantity can therefore offset one another in order to achieve the same fitness benefit when $\phi \leq \phi_{stable}$. This is not surprising given that, in the study system, large swaths of historical low-elevation rearing habitat for fry are inaccessible or have been destroyed[50], and smolts consistently face unnaturally low flows when migrating to the ocean[24–26].

Under laboratory conditions where $O_2$ was not limiting, FAS was above $\phi_{stable}$ for some individuals up to 22 °C in fry and 24 °C in smolts. In alignment with theory[15,22], this suggests that the realized niche, as estimated by $\phi_{stable}$, fell within the fundamental niche, as estimated by FAS, across the tolerated temperature range. However, FAS was only above $\phi_{stable}$ for the majority of fish tested below 14 °C in fry and 19 °C in smolts (Fig. 3). This suggests that, without rapid phenotypic plasticity (e.g., ref. 47), fundamental limitations may prevent some portion of captivity-raised Chinook salmon populations from exploiting the full realized niche when introduced into the wild under relatively warm conditions.

**Non-native predators may have been detrimental to prey fitness when aerobic scope was limited**

Using spatiotemporal models of $\phi$, we investigated if and how lifestage-specific physiology and phenology influenced vulnerability to variability in AS. The spatial pattern of $\phi$ was seasonally consistent for both lifestages, and the phenology of rearing and migration against this pattern highlighted regions of the Delta where AS-specific protection and restoration measures might be focused (Supplementary Fig. 2). Across the Delta, we found that the seasonal cycle of $\phi$, which peaked in winter and troughed in summer, was more pronounced in smolts (GAM: $R^2 = 0.84$, df = 85, residual df = 13,837) (Supplementary Equation 5) than fry (GAM: $R^2 = 0.64$, df = 90, residual df = 13,832) (Supplementary Equation 6) (Fig. 4A). In addition, fry typically reared in the Delta during peaks in $\phi$—by contrast, smolts typically migrated during periods when $\phi$ was rapidly changing (Fig. 4A). Thus, the metabolic traits of smolts (Table 1) placed this lifestage at greater risk of metabolic catastrophe depending on migration phenology. On the other hand, while $\phi$ rarely dropped below $\phi_{crit}$ for fry, it also infrequently exceeded $\phi_{stable}$ by a great margin. The metabolic traits and habitat use phenology of fry may therefore have largely avoided metabolic catastrophe at the expense of a consistently low metabolic safety margin.

The seasonal pattern of $\phi$ further contextualized the fitness relevance of AS in the wild. Spatiotemporal models predicted that daily $\phi$ averaged across the Delta (Fig. 4A) never dipped below $\phi_{crit}$ for rearing when fry were present, but dropped below $\phi_{crit}$ for migration during 18.4% of the days when smolts were present. Days when $\phi$ was at or between $\phi_{crit}$ and $\phi_{stable}$ when fry and smolts were present were similarly infrequent, only occurring during 17.6 and 15.3% of the seasonal window for rearing and migration, respectively. The most common predicted daily $\phi$ category for both rearing and migration was $\phi$ being above $\phi_{stable}$, which occurred during 82.4 and 66.2% of days when fry and smolts were present, respectively. Acknowledging that these results are generalized over spatial (Supplementary Fig. 2) and temporal (Fig. 4A) complexity, the magnitude and duration of $\phi$ fluctuations at present in the Delta suggest that AS may similarly limit and facilitate migration, but largely facilitate rearing. However, AS was often high enough to have no measurable influence on the quantified aspects of ecological fitness.

Predation by non-native warmwater fish (e.g., largemouth bass, *Micropterus salmoides*) is hypothesized to be a major factor contributing to the mortality of rearing and migrating juvenile salmonids in California[37,39]. However, predation in the field is mainly understood in terms of predator activity—we therefore investigated the potential contribution of both predator and prey AS to the outcome of their interactions. Accordingly, we first determined the metabolic traits of largemouth bass (LR: $R^2 = 0.71$, $F_{1,33} = 84.9$, $p < 0.001$) (Supplementary Equation 7), and revealed that this warmwater predator had greater hypoxia tolerance ($\pm 95\%$ CI) ($A = 21.3 \pm 2.16$) and temperature sensitivity ($-E = 0.50 \pm 0.05$) (Supplementary Table 3) than its cold-water prey (Table 1). The spatiotemporal model of largemouth bass $\phi$ in the Delta (GAM: $R^2 = 0.89$, df = 81, residual df = 13841) (Supplementary Equation 8) therefore exhibited a seasonal cycle that was more pronounced than that of both juvenile Chinook salmon lifestages (Fig. 4A). Higher temperature sensitivities are often equated to AS detriments under warmer conditions, but largemouth bass exhibited compensatory hypoxia tolerance, which fell in the 80% quantile of evaluated

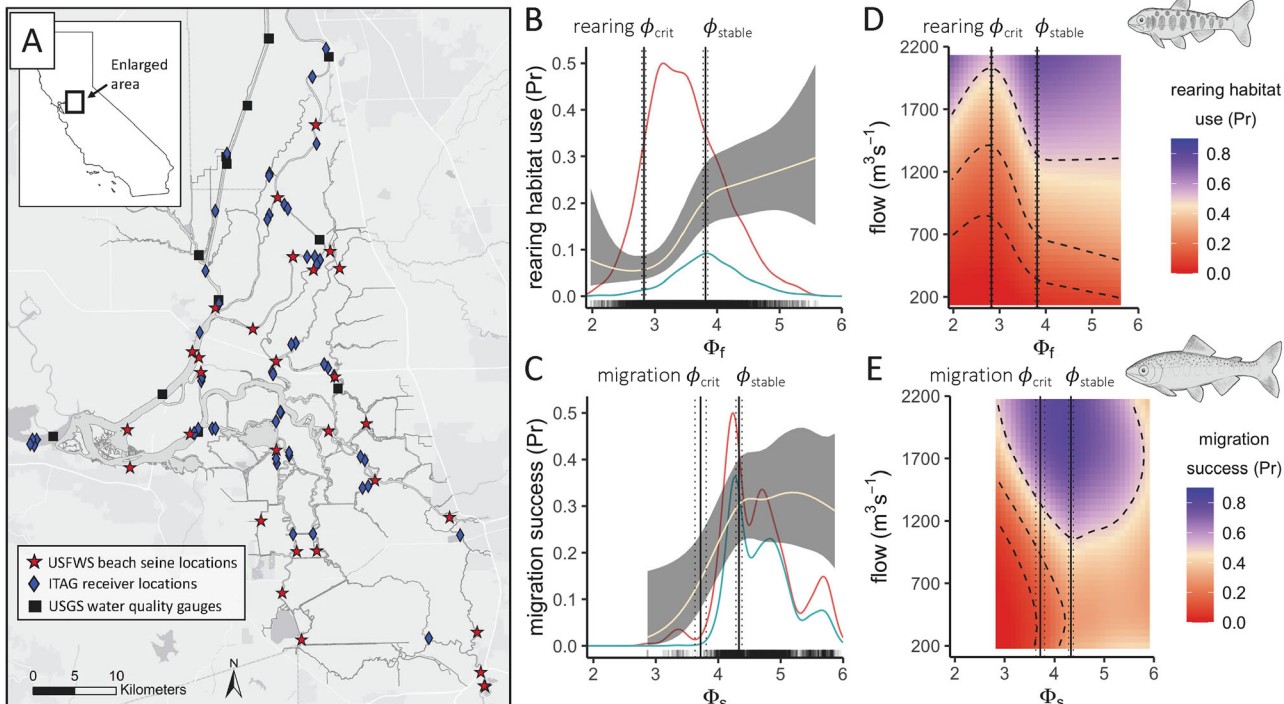

**Fig. 2 | Aerobic scope governed rearing and migration success only under specific conditions in juvenile Chinook salmon. A** Map of the Delta study area and study sites. Red stars show sampling locations where rearing Chinook salmon fry were repeatedly encountered from 2011 to 2022[35], blue diamonds show locations of acoustic receivers for migrating Chinook salmon smolts from 2019 to 2022[34], and black squares show locations where water temperature, dissolved $O_2$, and salinity data were recorded from 2019 to 2022[36]. (**B, C**) $\phi$ vs. ecological fitness, defined by **B** fry rearing habitat use or **C** smolt migration success probability. Light lines and gray ribbons respectively show predictions and 95% CIs for how probabilities changed across the full range of sampled $\phi$, while averaging over other parameters (see Supplementary Equations 3 and 4). The red and blue lines, respectively, show how the density of "0" and "1" observations in each dataset aligned with $\phi$, and are presented on a 0 to 0.5 scale. Black vertical solid lines show $\phi_{crit}$ and $\phi_{stable}$ for each behavior, and the respective 95% CIs are shown with dashed lines. **D, E** Gradients show predicted probabilities of **D** fry rearing habitat use and **E** smolt migration success under potential flow by $\phi$ combinations. Predictions are only shown across 0.01–0.99 flow quantiles due to the high uncertainty associated with rare flow conditions. Dashed lines show 0.10, 0.25, and 0.5 contours. $\phi_{crit}$ and $\phi_{stable}$, determined across the full range of flows, are shown using the same scheme as in (**B, C**). $\phi_f$ = fry-specific $\phi$, and $\phi_s$ = smolt-specific $\phi$, both averaged across populations (i.e., determined using LMER parameters in Table 1).

ectothermic taxa[7]. As a result, largemouth bass always had an AS surplus over juvenile Chinook salmon: on average, $\phi$ was 56% and 43% greater for the predator than fry and smolts, respectively.

To investigate if and how an AS advantage was relevant in predator–prey interactions, we modeled a metric of predator activity with respect to predator $\phi$: the probability that largemouth bass predated live, tethered juvenile Chinook salmon (GAMM: $R^2 = 0.24$, df = 21, residual df = 2256) (Supplementary Equation 9) in the interior Delta (Supplementary Fig. 3). Notably, $\phi$ GAMMs had lower BIC scores than temperature GAMMs of the same form (Supplementary Equation 10), suggesting that predator AS explained more variation in predation probability than temperature (Supplementary Table 4). This is not surprising given that largemouth bass are ambush foragers, and that digestion can require relatively large $O_2$ costs for ambush foraging fish[51]. Seasonal patterns of $\phi$-based predation probability showed an inverse pattern against the $\phi$-based ecological fitness of prey (Fig. 4B), suggesting that predation likely contributed to observed Chinook salmon fitness detriments. However, predation did not occur when largemouth bass had the largest AS advantage (Fig. 4A), likely because cold temperatures under such conditions limited metabolism[32] and predation activity[37,39]. Rather, predation probability was highest when the predator's AS advantage over prey was relatively small (Fig. 4A, B), which occurred when AS<$\phi_{stable}$ for prey.

In captivity, the ability of juvenile Chinook salmon to undertake repeat swimming bursts is a more important determinant of predation risk than the aerobic advantage of largemouth bass (McInturf et al 2022). While escape behaviors are typically anaerobic, the resulting aerobic debt must be paid off using energy and $O_2$. Even small impairments to prey AS

could therefore be relevant for the outcome of predator–prey interactions[13,15], especially when repeat encounters take place under aerobically limiting conditions, as occurs in the Delta[37,39] (Fig. 4A, B). In this context, our results suggest that prey AS impairments in predator–prey interactions could be involved in the putative association between AS and fitness in the wild.

### Applicability of aerobic scope-based management actions in the study system

Temperature covaries with many environmental factors such as food availability and the activity of predators[37,39] and parasites[52]; it is therefore a useful but confounded metric of water quality. $\phi$, on the other hand, specifically quantifies the AS of a given habitat and therefore directly assesses the impact of temperature and $O_2$ on physiological performance. Similar to studies on the long-term distributions of marine species[7,46,53], temperature offered no explanatory power over $\phi$ in determining the successful migration of captivity-raised smolts, as Bayesian information criterion (BIC) scores were equivalent for migration GAMs fit to $\phi$ or temperature (Supplementary Equations 4 and 11, respectively). This suggests that AS, in particular, plays a relatively large role in the well-documented association between smolt migration success and temperature in the Delta (e.g., refs. 44,54). However, the rearing GAMM with temperature (Supplementary Equation 12) instead of $\phi$ (Supplementary Equation 3) had a lower BIC score, indicating a better fit to the data (Supplementary Table 2). Thus, while AS plays some role in determining rearing habitat use by fry, it alone does not encompass the impact of temperature. This suggests that physiological and ecological differences between smolts and fry may make migrating

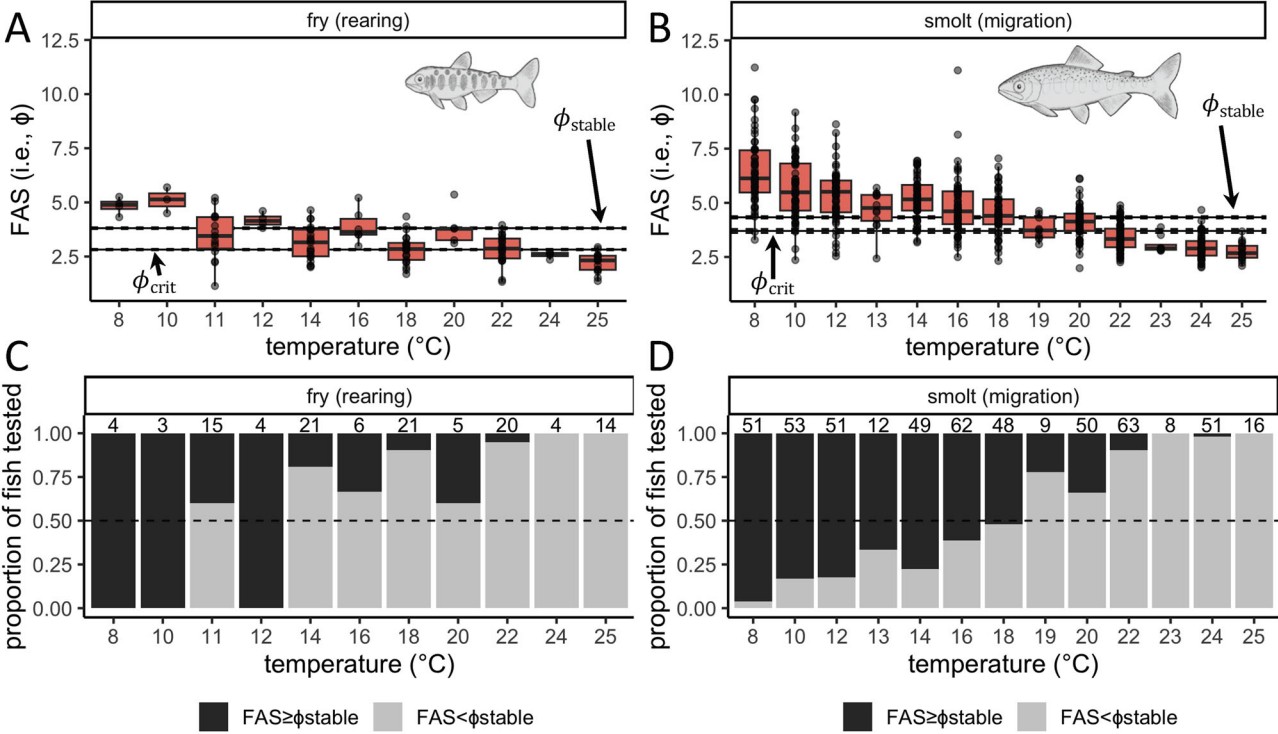

**Fig. 3 | The realized niche of juvenile Chinook salmon fell within the fundamental aerobic scope niche across the tolerated temperature range. A, B** FAS with respect to temperature for **A** fry and **B** smolt lifestages under experimental conditions where $O_2$ was not limiting. Stable and critical values of ϕ are shown with solid horizontal lines, and the respective 95% CIs are shown with dashed lines. **C, D** The proportion of fish tested (in **A, B**) that exhibited FAS above or below $\phi_{stable}$. Numbers above bars show the sample size for each treatment, and dashed horizontal lines show a proportion of 0.5.

smolts more vulnerable to AS limitations, but also more receptive to AS surpluses. But at a broader level, that fry are sensitive to other temperature-associated factors aside from AS supports the notion that AS is not a silver bullet—its utility as a metric of water quality is clearly circumstantial[14], and in this case, dependent on lifestage.

The probability of successful through-Delta migration by Chinook salmon smolts is widely recognized to deteriorate once Delta temperatures climb above 12 °C, becoming exceptionally improbable by 20 °C (e.g., refs. 44,54). Given that temperature covaries with numerous environmental factors, the exact mechanism of this relationship remains unresolved. The search for a mechanistic understanding has largely centered around smolt physiological performance and predator activity. Interestingly, performance, as assessed via thermal tolerance assays under laboratory conditions, typically increases from 12 to 20 °C[20,31–33]. Predator activity also increases over this temperature range[37,39], suggesting that predator activity, not prey performance, is the mechanism by which temperature impacts the outcome of migration[54]. If this is true, then predator control (e.g., ref. 39), not water quality improvement (e.g., ref. 30), would be the prescribed management tactic. However, given the specific goal of quantifying thermal tolerance, laboratory studies that measure juvenile salmon performance ensure that dissolved $O_2$ is as close to full air saturation as possible during experiments (e.g., refs. 20,31–33). By contrast, in aquatic environments such as the Delta, water naturally holds less $O_2$ as it warms, and can exhibit additional $O_2$ declines due to other environmental factors (e.g., respiration of submerged aquatic vegetation)[40]. We therefore quantified the synergistic impacts of temperature and $O_2$ on performance using AS (i.e., ϕ) and found that, if $O_2$ is sufficiently limiting, juvenile Chinook salmon performance can be meaningfully restricted from 12 to 20 °C (Fig. 5). While predators are undoubtedly a proximal cause of migrating smolt mortality, our results suggest that the ultimate cause is poor water quality—by restricting AS (i.e., $\phi \leq \phi_{stable}$), warm and low-$O_2$ conditions curb aerobic activity and recovery from bouts of anaerobic activity, making migrating smolts easier targets for predators that always have an aerobic advantage (Fig. 4A)[40]. Thus, prey

performance is indeed a key mechanism by which temperature impacts the outcome of migration, and water quality improvement would therefore be a logical management tactic.

Taken together, our results imply distinct AS-based management actions to respectively increase the success of fry rearing and smolt migration when $\phi \leq \phi_{stable}$. Given a relatively low hypoxia tolerance and temperature sensitivity (Table 1) and longer-term residence in the Delta (Fig. 4A), rearing fry may respond to strategies that result in sustained increases in $O_2$ (Fig. 5A), such as nutrient control, submerged aquatic vegetation removal, channel shallowing, and physical re-aeration[40,55,56]. In contrast, migrating smolts had greater hypoxia tolerance and temperature sensitivity than fry (Table 1), and were only temporary inhabitants of the Delta (Fig. 4A); smolts may therefore respond to more drastic but ephemeral actions centered around temperature regulation (Fig. 5B), such as cooling flow deliveries[30], or the creation of localized thermal refugia[57] during migration peaks. Temperature regulation could have the added benefit of limiting mortality due to predation, either by reducing predator activity[37,39], or by enhancing prey aerobic recovery between anaerobic bouts of predator escape[32]. However, even if cold water supply is limited, our models suggest that any increase in flow could offset AS-related fitness detriments when $\phi \leq \phi_{stable}$ (Fig. 2D, E).

Given the constraints of available data, two key limitations of our study must be acknowledged. First, our metric of fry rearing did not directly measure growth or survival, but assumed that both were related to the availability of suitable rearing habitat (e.g., ref. 48). Second, environmental data were synchronized with smolt migration attempts post-hoc using water quality gauges, and we used average values across the duration of migration as independent variables (e.g., ref. 44). For the former, future research might use advances in the miniaturization of acoustic tags to explore how the growth and survival of fry is related to habitat suitability on an individual basis. For the latter, future research might, for example, use temperature-logging acoustic tags or explore different methods of summarizing environmental conditions during migration (e.g., minima or duration at minima instead of averages).

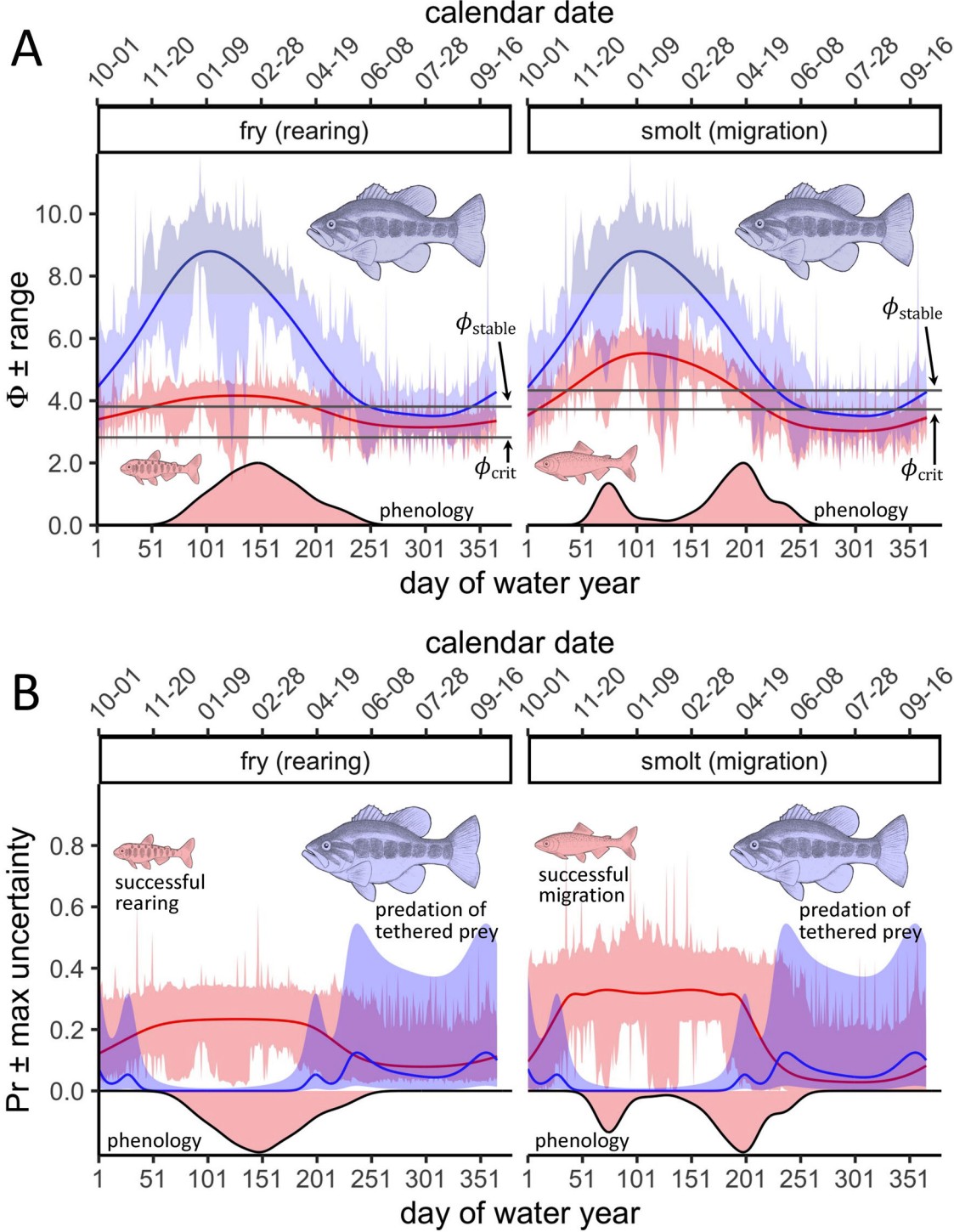

**Fig. 4 | Despite a continuous aerobic scope advantage, largemouth bass predation activity only co-occurred with fitness declines when juvenile Chinook salmon were aerobically constrained. A** Ribbons show the range of daily ɸ determined from temperature and $O_2$ measurements collected at monitoring locations within the Delta (Fig. 2A) over three water years spanning critically dry to wet classification (2019–2021). Lines are the 95% CIs of spatiotemporal ɸ GAMs predicted at an average location within the study system (see Supplementary Equations 5, 6, and 8). Red = fry or smolt ɸ, blue = largemouth bass ɸ. Upper and lower gray horizontal lines respectively show ɸstable and ɸcrit for rearing or migration. Density plots, which are presented on a 0 to 2 scale, show phenological patterns of fry habitat use and smolt through-delta migration attempts over the same time period as plotted ɸ. **B** Lines show predictions of fry or smolt fitness and largemouth bass predation activity based on seasonal patterns of ɸ in (**A**) while averaging over other parameters (see Supplementary Equations 3, 4, and 9). Ribbons show maximum quantifiable uncertainty: upper estimates are the upper 95% CI of the rearing ɸ GAMM, migration ɸ GAM, or predation ɸ GAMM predicted using the upper range of ɸ in (**A**), while lower estimates are the lower 95% CI of ɸ models predicted at the lower range of ɸ. Red = fry or smolt fitness, blue = largemouth bass predation. Density plots are the same as in (**A**), but inverted for clarity and presented on a 0 to −0.2 scale.

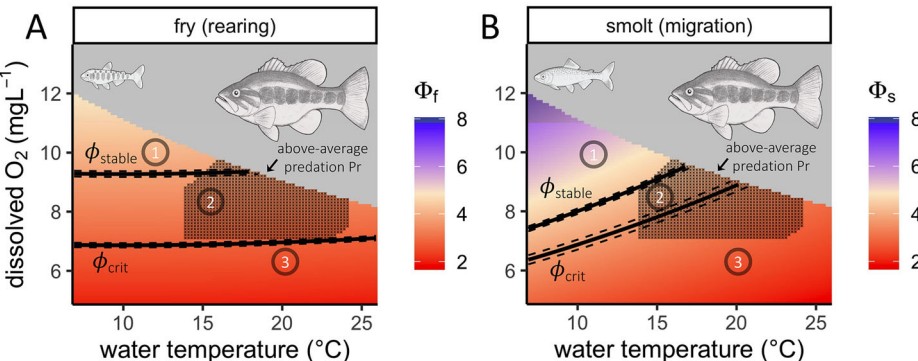

**Fig. 5 | Metabolic traits in an ecological context imply distinct management actions to increase the success of Chinook salmon rearing and migration when aerobic scope is relevant for ecological fitness.** Predicted effects of potential temperature and $O_2$ scenarios on **A** rearing fry and **B** migrating smolts constrained to the range of environmental conditions in both datasets. φ quantifies the synergistic impact of temperature and $O_2$ on aerobic scope ($\phi_f$ = fry-specific φ, and $\phi_s$ = smolt-specific φ). Flow does not change ecological thresholds of φ (i.e., $\phi_{crit}$ and $\phi_{stable}$), only the baseline probability of fry rearing habitat use and smolt migration success (Fig. 2D, E). Regions where simulated conditions would result in $O_2 > 21$ kPa (air saturation of $O_2$ at sea level) are shaded gray. Black solid lines show $\phi_{crit}$ and $\phi_{stable}$ for rearing and migration (Fig. 2B, C), and the respective 95% CIs are shown with dashed lines. Small black lines show conditions where largemouth bass predation probability was predicted to be above average (mean, lower 95% confidence limit [CL], and upper 95% CL = 0.013, 0.0018, and 0.089, respectively), constrained to the range of environmental conditions within this dataset. Circled numbers represent potential temperature and $O_2$ scenarios corresponding to the following general guidance: (1) conditions are above $\phi_{stable}$ and predation is below average-temperature reductions and $O_2$ increases would have little impact on rearing or migration but increases in flow might be beneficial; (2) conditions are between $\phi_{crit}$ and $\phi_{stable}$ and predation is above average—relatively small temperature reductions and $O_2$ increases would result in relatively large benefits to migration and rearing, as would increases in flow, in part by mitigating predation threat; (3) conditions are below $\phi_{crit}$ and predation is below average-temperature reductions and $O_2$ increases might be beneficial for rearing and migration (as might increases in flow), but alterations would likely have to be relatively large to see any benefit. Note that: $\phi_{crit}$ and $\phi_{stable}$ contours for fry rearing change little with temperature but drastically with $O_2$, and $\phi_{crit}$ and $\phi_{stable}$ contours for smolt migration change similarly with both temperature and $O_2$. Thus, when appropriate, temperature reductions and $O_2$ increases are both viable strategies to improve smolt migration success, while $O_2$ increases would be most impactful for fry rearing.

While the impact of reservoir releases on downstream temperatures are becoming increasingly resolved in the study system[29,30], AS-based management actions in the Delta are additionally limited by a dearth of information on how dissolved $O_2$ could be increased over short and long timescales. Thus, there is a need to test the efficacy of relevant strategies that have been applied elsewhere[55,56]. While our study provides some guidance on where AS might benefit from habitat restoration (Supplementary Fig. 2), more detailed and systematic surveys of temperature and $O_2$ in the Delta would greatly assist in such efforts. Similarly, the conservation of Chinook salmon in California's Sacramento-San Joaquin watershed would benefit from an understanding of how AS is relevant for the success of mature adult migrations to upstream spawning grounds.

## Conclusions

As Earth's climate warms and precipitation becomes more variable[58,59], intensified anthropogenic modification of freshwater ecosystems and ensuing management of their resources are resulting in drastic declines of ecologically, economically, and culturally important taxa, including Chinook salmon (e.g., refs. 24,25,60). Given that further anthropogenic interventions are mandated to prevent extinctions and recover imperiled taxa (e.g., ref. 61), it is critical to guide these efforts by resolving ecological thresholds of water quality and the relevant mechanistic underpinnings[62]. Water quality targets are often based on the results of laboratory studies alone (e.g., ref. 63)—given the frequent discrepancy between performance in captivity vs. the wild[64], we interrogated the utility of AS in facilitating the recovery of imperiled salmon populations by linking φ to fitness in the wild alongside other important environmental factors.

There are three general patterns by which AS is hypothesized to be relevant for fitness in the wild. If AS governs most processes underlying organismal performance (i.e., oxygen and capacity limitation of thermal tolerance [65]), then AS could constitute a (1) fitness-limiting barrier[5,10] or a (2) fitness-facilitating avenue[13], depending on patterns of natural selection. However, if AS instead comprises one of many aspects of organismal performance rather than governing the entire suite (i.e., multiple performances-multiple optima[14]), then AS might only be (3) circumstantially important for fitness under natural settings[14]. As observed at the warmwater limits to the distributions of marine ectotherms[10], we identified $\phi_{crit}$ values for both rearing and migration bottlenecks in Chinook salmon, supporting hypothesis 1. However, our results clarify that any resulting population-level impact depends on the fitness-associated behavior in question. In addition to $\phi_{crit}$, we found evidence of $\phi_{stable}$, the point above which increased AS conferred no additional fitness benefit. We posit that conditions at and between $\phi_{crit}$ and $\phi_{stable}$ represent the ecological threshold[11,12] where AS constitutes a fitness-facilitating avenue in viable habitats, supporting hypothesis 2. Finally, we found that conditions in our study system were often such that AS had no discernable impact on fitness (i.e., AS>$\phi_{stable}$), or that other parameters (i.e., temperature and its associated threats) were more important for fitness than φ, supporting hypothesis 3. Thus, rather than a contrast between limiting, facilitating, and circumstantial, our results suggest that AS can circumstantially both limit and facilitate fitness in the wild.

If organisms have the greatest potential to leverage intrinsic metabolic traits toward an ecological advantage when additional AS becomes obtainable between $\phi_{crit}$ and $\phi_{stable}$, then $\phi_{stable}$ may serve as a more conservative AS benchmark than $\phi_{crit}$ to prevent extirpations or extinctions and recover imperiled populations under a changing climate. As highlighted by our results, strategies to mitigate fitness costs ultimately attributable to AS, which could include those proximally resulting from predation, should be tailored to specific metabolic traits and habitat use phenologies. However, altering φ via temperature and $O_2$ interventions may not always be the most prudent course of action in combating reduced habitat quality. For example, we found that strategically expanding habitat quantity (e.g., via flow deliveries)[24–26] would bolster the fitness of Chinook salmon even under AS-specific limitations in habitat quality.

While other study systems might reveal different patterns than reported here, datasets that could evaluate the ecological importance of AS are rare[3,14,15]. Prior to our study, the most relevant research to date has been conducted in marine environments, with expansive taxonomic coverage but limited ecological resolution[10]. In contrast, freshwater environments offer relevant spatiotemporal resolution[16,66] that is not yet obtainable in the ocean, and also contain organisms that cannot readily shift distributions in response to a changing climate, unlike many marine organisms (e.g., ref. 8). Accordingly,

the interpretive limitations of ϕ and other AS-based metrics of water quality caused by untested assumptions (e.g., that the adaptive capacity of metabolic traits is outpaced by climate change)[3], stand to further benefit from unexplored ecological patterns in freshwater environments.

## Methods

### Metabolic trait determination

All analyses and visualizations were conducted in R version 4.1.0[67]. To determine the metabolic traits of juvenile Chinook salmon, we used data from published intermittent respirometry experiments from the same research group that measured standard metabolic rate (SMR) and maximum metabolic rate (MMR) of captivity-raised fry and smolts comprising all populations[20,31–33]. These studies were performed over several years and had slight methodological differences (e.g., respirometers, measurement cycle durations, and fasting durations), which are discussed in more detail below. SMR, MMR, and resultant aerobic scope (AS) can be repeatable traits[68] that are posited to allow for a more nuanced assessment of how species interact with their environment[69].

Depending on fish size (mass range of tested fry and smolts was 0.9–7.0 g [$n = 117$] and 13.2–40.9 g [$n = 523$], respectively), intermittent respirometry was conducted in either 1.5 or 5 L respirometers. Following[45], SMR values were calculated using metabolic rates gathered on fasted fish (24–48 h of fasting depending on temperature) during an overnight period[20,31–33]. SMR was determined over measurement periods that ranged from 12 to 24 h in these studies, which is considered relatively short[45]. Any inflation of SMR attributable to short measurement periods could be influential when determining factorial AS (FAS)[70]. However, fish were monitored during all respirometry trials using infrared cameras; outside of small fin movements to maintain position in respirometers[45], fish movement was excluded from SMR values. Thus, the values used were considered a decent approximation of SMR and, therefore, unlikely to impact FAS, nor the overall results of our study.

MMR values were determined using either exhaustive chase (54 of the fall-population smolts)[31] or forced swimming (the remaining 586 fish)[20,32,33] procedures. During exhaustive chase, which followed a published protocol[71,72], individuals were manually chased to exhaustion until they no longer responded to caudal fin contact, and MMR was measured in a static respirometer. During forced swimming, which followed a modified $U_{crit}$ protocol[73,74], individuals were required to swim against a stepwise increasing current in a swim tunnel respirometer until exhaustion. Exhaustive chase and forced swimming protocols have been shown to yield equivalent MMR values[75] (but see ref. [76]), and we found that the MMR method did not affect metabolic trait determination (Supplementary Figs. 4–6).

Our metabolic data sources quantified SMR and MMR across a range of ecologically relevant acclimation (11–20 °C) and testing temperatures (8–25 °C). While acclimation temperature can affect metabolic rates, we found that it did not affect metabolic trait determination (Supplementary Table 5). We converted paired measurements of SMR and MMR to $O_2$crit using the following relationship (Eq. 2), which has been validated for diverse taxa, including salmonids[13].

$$\frac{MMR}{SMR} = \frac{O_2 crit\ of\ MMR}{O_2 crit\ of\ SMR} \qquad (2)$$

SMR and MMR were measured at as close to 21 kPa (air saturation of $O_2$ at sea level) as possible in all intermittent respirometry experiments[20,31–33], which is standard practice for salmonids and other normoxic species[13]. We therefore assumed that $O_2$crit of MMR was equal to 21 kPa and then solved for $O_2$crit of SMR. While some salmonids have been shown to achieve higher MMR at hyperoxia than normoxia[77], $O_2$crit of MMR has been experimentally shown to be 21 kPa in other salmonids[78,79], validating our assumption to apply normoxic constraints. Moreover, normoxia is far more prevalent than hyperoxia in the study system.

To determine the metabolic traits $A$ and $-E$, we first standardized mass-specific $O_2$crit to a common temperature (the average in our data, 16.6 °C) using the association between the natural logarithm (ln) of mass-specific

$O_2$crit and inverse temperature (linear regression [LR]: $R^2 = 0.07$, $F_{1,638} = 48.8$, $p < 0.001$). Next, we standardized the data to a common mass (the average in our data, 0.02 kg) using the association between the ln of temperature-standardized, mass-specific $O_2$crit and the ln of mass (Supplementary Fig. 1). The resulting normalization constant (β) and scaling coefficient (α) were 1.16 ($t = 23.4$, $p < 0.001$) and $-1.12$ ($t = -96.0$, $p < 0.001$), respectively. We did not test for population-specific mass scaling because not all populations were represented with a sufficient size range in the data. Following[5], $A$ and $-E$ were respectively determined from the intercept and slope of the multiple regression (MR) associating inverse temperature and the ln of mass-standardized $O_2$crit (Fig. 1). The best supported MR (Supplementary Table 1 and Supplementary Equation 2) also included fixed effects of lifestage and population, and an interaction between inverse temperature and lifestage. To determine lifestage-specific metabolic traits averaged across populations, we fit a linear mixed-effects regression (LMER) with a by-population random intercept (Supplementary Equation 1) using the "lme4" package[80]. The "bootMer" function in this package was used to generate bootstrapped confidence intervals for LMER predictions that incorporated variance around the random intercept (Fig. 1). Both the MR and LMER were verified to have satisfactory diagnostics (Supplementary Figs. 5 and 6).

### Fitness benefits attributable to aerobic scope

The probability of Delta habitat use by wild rearing Chinook salmon fry (i.e., salmon presence or absence) was estimated from a US Fish and Wildlife Service monitoring dataset of beach seine catches and concurrently measured environmental conditions (water temperature, dissolved $O_2$, and salinity)[35]. Sampling locations outside of the Delta, and those where fry were caught in fewer than 5% of sampling events over the entire 12-year sampling duration (2011–2022), were removed (45% of locations removed), bringing the total number of locations where repeat sampling occurred to 15 (Fig. 2A). Sampling events with temperature, $O_2$, or salinity exceeding the upper or lower 0.005 quantiles of were also removed for quality control (3% of sampling events removed). We then calculated $O_2$ partial pressure using the "respirometry" package[81] for the remaining sampling events ($n = 5401$).

Catch was converted to the presence or absence of wild fry, which were identified as Chinook salmon that were unmarked and below the 0.05 length quantile of all marked hatchery smolts in the dataset (6.7 cm). We chose the size threshold to conservatively exclude captivity-origin fish. Length-at-date criteria[35] suggested that fry of all populations (fall, late fall, winter, and spring contributed to this analysis (Supplementary Fig. 7). Tidally filtered average daily discharge of the Sacramento River into the Delta (DAYFLOW) was obtained from the California Data Exchange Center and associated with each sampling event.

We used generalized additive mixed-effects models (GAMMs) implemented using the "mgcv" package[82] to understand how ϕ was associated with fry habitat use probability, and how its effect compared to that of flow and temperature. GAMMs were used to allow for non-parametric predictor-response associations, and to incorporate fixed and mixed effects. Since ϕ and temperature were highly correlated, leading to excessive concurvity (≥0.7) (the extent to which independent variables approximate one another's impact on the dependent variable, ranging from 0 to 1) when included in the same model, we assessed their effect using two separate models. Both used a binomial family with logit link function, and included a by-sampling location random intercept (15 locations where repeat sampling occurred) and smooth terms (cubic regression spline) for flow and ϕ or temperature (Supplementary Equations 3 and 12, respectively). To explore a hypothesized synergistic impact of water quantity and quality on rearing habitat use, a parametric interaction between the flow and ϕ or temperature smooth terms was additionally included. For both rearing GAMMs, 7 knots (k = 7) were sufficient to represent non-parametric associations given this yielded model convergence, a lack of concurvity, and satisfactory residual diagnostics (Supplementary Figs. 8 and 9), which we obtained using the "DHARMa" package[83].

Through-Delta migration success probability of Chinook salmon smolts (successful or unsuccessful migration) was estimated using acoustic telemetry studies from 2019 to 2022[34]. Thousands of acoustically tagged smolts were released in groups at various locations in the Sacramento River basin upstream of the Delta, where dozens of strategically positioned acoustic receivers recorded their movements (Fig. 2A and Supplementary Fig. 10). Given near-perfect detection efficiency at the downstream end of the Delta (mean = 1, range = 0.8–1 for all release groups), we included all smolts that were detected at any point from the upstream end to downstream of the Delta in this analysis ($n = 2240$). Through-Delta migration success was therefore represented by the detection of a tagged fish at acoustic receivers located at the downstream end of the Delta (near Benicia, California) or at ocean entry (near the Golden Gate Bridge). If smolts were not detected at these locations, this was interpreted as a migration failure.

Using timeseries of detections for each tagged smolt, as revealed by acoustic receivers positioned throughout the Delta (Fig. 2A and Supplementary Fig. 10), we determined that three general routes were taken during the study period: the Yolo Bypass, Sacramento River, and Interior Delta (Supplementary Fig. 10). We assumed that detections accurately represented smolt routing, and therefore assigned each smolt to the route it was detected within. However, if detections were missed (i.e., a smolt took a given route but was not detected within the route), this could have biased route assignment and, therefore, the environmental parameters associated with the migration attempt. Given that there was greater temporal vs. spatial variation of $\phi$ in the Delta (Fig. 4A and Supplementary Fig. 2), the impact of route assignment on the $\phi$ associated with each migration attempt was likely less than the timing of the migration attempt, which was quantified using receivers with near-perfect detection efficiency.

Water temperature, dissolved $O_2$, and salinity measurements collected by the US Geological Survey[36] were compiled using the "dataRetrieval" package[84] from 12 stations that generally covered the routes smolts took through the Delta (Fig. 2A and Supplementary Fig. 10). Periods of time where temperature or $O_2$ data were missing were excluded from analysis (7% excluded). If salinity data were missing for a period of time, average salinity at the station or across non-estuarine stations was used. From these data, we calculated $O_2$ partial pressure[81]. Given that the resolution of both the environmental and acoustic telemetry data were similar, we assumed that environmental conditions as recorded by USGS stations were adequately representative of conditions experienced by migrating smolts. However, it is possible that finer-scale movements could reveal that migrating smolts are experiencing conditions that are distinct from those recorded by USGS stations.

Each smolt was associated with the mean value of route-specific temperature and $O_2$ partial pressure for the time it was present, presumed to be present (if missing detections near upstream end), or would have been present (if missing detections near downstream end) in the Delta. Note that Yolo Bypass and Interior Delta routes eventually rejoin the Sacramento River (Supplementary Fig. 10). While averaged environmental conditions were used in this study, it is important to note that other summarizing methods (e.g., maxima or minima) could be relevant. Missing detections at the upstream and/or downstream end of the Delta were interpolated using the shortest river distance between the temporally closest detection and the missing location using the "riverdist" package[85], and the most precise average travel rate possible (release group by route, all fish by route, or overall average for all routes and fish). We made the necessary assumption that the travel rate of other smolts was representative of the travel rate of smolts with missing detections. Mean tidally filtered flow was also associated with each smolt.

Similar to a previous telemetry analysis where we leveraged near-perfect detection efficiency[44], generalized additive models (GAMs)[82] were used to assess predictor effects on through-Delta migration success probability of smolts. Compared to traditional mark-recapture models (which do incorporate detection efficiency), GAMs have greater flexibility for non-parametric predictor-response associations and more comprehensive residual diagnostics. This not only allowed us to examine hypothesized non-linearities, but also to ensure that all model assumptions were met (e.g.,

Supplementary Figs. 11 and 12). Smooth terms were included for flow, distance from release location, fish length, and $\phi$ or temperature (Supplementary Equations 4 and 11, respectively). Distance from release was included to account for any effects of river conditions or migratory demands prior to Delta entry. Differences in migration success attributable to fish size or swimming speed were encapsulated by the fish length predictor. As with the rearing analysis, we investigated a hypothesized synergistic impact of water quantity and quality on migration success using interactions between flow and $\phi$. A parametric interaction was included in the $\phi$ GAM only, as the corresponding interaction was not supported in the temperature GAM, given that it masked the effect of flow. For both migration GAMs, 7 knots ($k = 7$) were sufficient to represent non-parametric associations, and diagnostics were satisfactory (Supplementary Figs. 11 and 12)[83].

To determine $\phi$ breakpoints, we used the fits of the fry habitat use $\phi$ GAMM and smolt migration success $\phi$ GAM to predict probabilities across the full range of $\phi$ and flow while averaging over other parameters. Segmented regressions fit to log-odds predictions then estimated $\phi$ breakpoints for each lifestage using the "segmented" package[86]. Log-odds predictions were used to avoid spurious inflections resulting from transformation to probability, which forces predictions between 0 and 1. We found that two breakpoints, corresponding to $\phi_{crit}$ and $\phi_{stable}$, were sufficient to capture the major inflections of the curves and explained over 99% of the variation.

## Fitness detriments attributable to non-native predators

Published measurements of SMR and MMR for largemouth bass[32] were converted to $O_2$crit as specified for juvenile Chinook salmon. $A$ and $-E$ (Supplementary Table 3) were also determined in the same manner, except there was no need to test for lifestage or acclimation temperature effects, as all individuals were adults and a consistent acclimation temperature was used. Accordingly, temperature standardization (LR: $R^2 = 0.69$, $F_{1,33} = 77.7$, $p < 0.001$) and mass standardization (Supplementary Fig. 13) (LR: $R^2 = 0.12$, $F_{1,33} = 5.49$, $p = 0.03$) ($\beta = 2.07$ [$t = 23.4$, $p < 0.001$], $\alpha = -0.54$ [$t = -96.0$, $p = 0.03$]) were applied to standardize $O_2$crit to a common mass (the average, 0.23 kg), and then the temperature sensitivity of mass-standardized $O_2$crit was determined (Supplementary Fig. 14 and Supplementary Equation 7.

GAMs were used to assess spatiotemporal patterns of $\phi$ from measurements of daily average temperature, $O_2$, and salinity collected at monitoring locations within the study region. Data used for this analysis covered three water years spanning critically dry to wet classification (2019 to 2021). Separate models were fit for Chinook salmon fry (Supplementary Equation 5), smolts (Supplementary Equation 5), and largemouth bass adults (Supplementary Equation 8), as each had distinct metabolic traits, resulting in different $\phi$ under the same conditions. GAMs included smooth functions for latitude and longitude ($k = 15$), day of water year ($k = 9$), and an interaction between both smooth terms. We used a scaled-t family with an identity link function and obtained satisfactory diagnostics (Supplementary Figs. 15–17)[82].

To investigate the association between largemouth bass AS and juvenile Chinook salmon predation, we used data from published deployments of predation event recorders (PERs) in the interior Delta[37–40]. PERs are freely floating, GPS-enabled devices designed to evaluate predation rates of juvenile Chinook salmon by aquatic predators[38]. Live juvenile Chinook salmon were tethered to PERs and monitored for the entire deployment duration (mean ± 1 SD deployment duration = 44 ± 33 min) using an underwater camera in order to determine the identity of predators[38]. A successful predation event occurred when a largemouth bass removed the prey from the PER. All predation events by other species, or predation events where the predator was not identifiable, were excluded from this dataset. Thus, no predation occurred when the juvenile salmon was not predated by a largemouth bass, nor any other predator, for the entire PER deployment.

A total of ($n = 2277$) PER deployments were used for this analysis (Supplementary Fig. 3). 1567 occurred in the San Joaquin River from 2014 to 2015[38,39]. Another 158 deployments occurred in the San Joaquin River, Middle River, and Old River in 2017[37]. These deployments were conducted with stationary PERs (sPERs), which were an anchored version of the freely floating model. Finally, 552 deployments occurred in the Middle River in

2022[40]. These deployments were conducted with pole PERs (pPERs), a shore-based version of sPERs where juvenile salmon were tethered to the end of a telescoping pole that was anchored to the bank. All PER deployments were synchronized with hydrographic conditions collected concurrently at each study site, including water temperature, salinity, and dissolved $O_2$. Other predation-related variables, such as time to night[38] and distance from shore[37] were also determined for the duration of deployments.

GAMMs were used to relate the probability of juvenile Chinook salmon predation by largemouth bass to largemouth bass $\phi$. As with rearing and migration analyses, we examined how the explanatory power of temperature compared with $\phi$ using a separate temperature model of the same form. All models used a binomial family with logit link function, and included by-method (PER, sPER, or pPER) and by-study site (30 locations where repeat sampling occurred) random intercepts, smooth terms for $\phi$ or temperature (averaged over deployment) (Supplementary Equations 9 and 10, respectively), and a parametric term for deployment duration (z-scored within method and study site). We also tested the effect of including median time to night and distance from shore (both z-scored within method and study site) by fitting models that additionally included these variables as smooth terms. Seven knots ($k = 7$) were sufficient to represent non-parametric associations, and the most parsimonious GAMM was the $\phi$ GAMM without median time to night and distance from shore (Supplementary Table 4). Given satisfactory diagnostics (Supplementary Fig. 18)[83], this was selected as the final model.

### Ethics approval for animal studies
No ethics approval was necessary given that this manuscript involved no new data collection.

### Reporting summary
Further information on research design is available in the Nature Portfolio Reporting Summary linked to this article.

## Data availability
All data supporting the findings (Supplementary Data 1–7) are deposited on Dryad (ref. 87), and can be downloaded using the following link: https://doi.org/10.5061/dryad.kprr4xhdr

## Code availability
All R code supporting the findings are deposited on Zenodo (accompanying data in ref. 87), and can be downloaded using the following link: https://doi.org/10.5281/zenodo.13774802.

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

## Acknowledgements
We thank Alyssa FitzGerald for providing thorough feedback that greatly improved the manuscript; Eric Danner, Steven Lindley, and Nate Mantua also provided insightful reviews. We are grateful to the many people and agencies who contributed to the publicly available datasets used in this study. This work was funded by the US Bureau of Reclamation grant agreements R18AP00136 and R21AC10455 (to C.J.M.); A.G.M. received support from the Cooperative Institute for Climate, Ocean, & Ecosystem Studies under NOAA Cooperative Agreement NA20OAR4320271, Contribution No. 2024-1332 (A.G.M.). The scientific results and conclusions, as well as any views or opinions expressed herein, are those of the author(s) and do not necessarily reflect those of NOAA or the Department of Commerce.

## Author contributions
Conceptualization: B.P.B., B.M.L., K.W.Z., V.K.L., G.T.K., and C.J.M.; Data curation: B.P.B., B.M.L., K.W.Z., V.K.L., A.G.M., D.E.C., N.A.F., and C.J.M.; Formal analysis: B.P.B. and C.J.M.; Funding acquisition: N.A.F. and C.J.M.; Investigation: B.P.B., B.M.L., K.W.Z., V.K.L., A.G.M., and C.J.M.; Methodology: B.P.B., B.M.L., K.W.Z., V.K.L., A.G.M., G.T.K., D.E.C., N.A.F., and C.J.M.; Project administration: B.P.B. and G.T.K.; Resources: N.A.F. and C.J.M.; Supervision: N.A.F. and C.J.M.; Validation: B.P.B.; Visualization: B.P.B.; Writing—original draft: B.P.B.; Writing—review and editing: B.P.B., B.M.L., K.W.Z., V.K.L., A.G.M., G.T.K., D.E.C., N.A.F., and C.J.M.

## Competing interests
The authors declare no competing interests.
