## [Transparent Peer Review file · Communications Biology]

Linking aerobic scope to fitness in the wild reveals potential opportunities to help recover imperiled salmon populations

Corresponding Author: Dr Benjamin Burford

Version 1:

Reviewer comments:

Reviewer #1

(Remarks to the Author)

Many empirical studies identify that aerobic metabolic capacity (Aerobic scope; AS) is commonly constrained at high temperatures and low environmental oxygen levels. However, the extent to which such constraints dictate performance of fitness linked traits, habitat use, and habitat suitability in nature is challenging to assess. To help address this challenge the present study integrates several existing data sources to model the association of the predicted aerobic capacity of juvenile chinook salmon with their habitat use, survival, and predation risk by a non-native predator. The models indicate that below a certain AS further environmentally driven reductions in AS are associated with reduced habitat use and migration success, and increased probability of predation. As with other reviewers I found the study to be a highly creative, and generally impressive undertaking.

The authors provided extensive rebuttals to prior reviews, however there were a few issues raised previously or associated with those comments that in my assessment were not adequately addressed.

General comments:

At 8 paragraphs and ~1300 words I found the introduction to be unnecessarily long and that it did not follow an obvious flow. As with other reviewers, with the number of concepts the manuscript tries to address in the introduction I found it difficult to follow or link together the core rationale. I think much of background on AS could be more direct. In particular, the logic in the paragraph from L35-46 was not obvious to me, nor is it clear specifically how this study would address this proposed dichotomy. The objective that immediately follows this paragraph does not seem to address the knowledge gap raised.

I also think the introduction somewhat misinterprets common views among experimental biologists about the likely importance of AS in an ecological context. I can't think of any that would argue that AS governs all aspects of organismal performance (157-58). Most would agree that the extent to which AS influences fitness is context dependent in natural settings, varying with both environment and life history demands (e.g., feeding vs upriver migration)(e.g., Farrell 2016). Some material specific to chinook salmon can also be greatly condensed and specifics can be raised in the methods or discussion where relevant.

The other issue raised by Reviewer 3 that I thought was not adequately addressed was related to methods used in the assessments of metabolic rates. The authors make several assertions about metabolic trait assessments that are not accurate. In particular, based on its definition, 'standard metabolic rate' has specific measurement criteria that were not met, and/or not standardized in the studies from which 'SMR' was taken from. The authors of those studies are clear to state that their measurements are referred to as RMR, not SMR, because they don't meet those criteria. In general, the differences among the studies may not impact the overall conclusions but they need to be addressed. I elaborate on these below.

Specific comments:

1) L88-92: This sentence could be shortened/split for clarity.

2) L98-101: simplify use of 'returning' to improve readability

3) L431: This information is intended to rebut a comment from Reviewer 3 regarding difficulties in comparing absolute values across studies. However, despite being from the group, the cited studies use quite different methods including different types of respirometers (swimming vs. static), measurement cycle durations, fasting durations, and measurements were performed over several years. In the end, the current investigation may be robust to these differences, but it cannot be implied that these studies used highly comparable methods. These methods are also not standard in the field as implied by referring to them as "established".

4) L 435: SMR, MMR, and Aerobic scope can be repeatable in some contexts, but it is not a rule as stated here.

Norin, T., & Malte, H. (2011). Repeatability of standard metabolic rate, active metabolic rate and aerobic scope in young

brown trout during a period of moderate food availability. *Journal of Experimental Biology*, 214(10), 1668-1675.)

5) L441: Fasting durations were as short as 24 hours

6) L442: Mo₂ measurement periods for RMR or SMR varied between 12 and 24 hours across studies to the point that the authors go out of their way to highlight that they use the term 'RMR' in 3 out of 4 of these studies rather than SMR because their approach was not suitable to reliably/confidently estimate SMR. Inflation of SMR may be particularly consequential when being used to calculate FAS

Halsey, L. G., Killen, S. S., Clark, T. D., & Norin, T. (2018). Exploring key issues of aerobic scope interpretation in ectotherms: absolute versus factorial. *Reviews in Fish Biology and Fisheries*, 28(2), 405-415.

7) L 451: These swimming protocols differ from those tested in the cited validation studies and there are other validation studies that show the similarity of these assessments varies greatly with analytical methods:

Little, A. G., Dressler, T., Kraskura, K., Hardison, E., Hendriks, B., Prystay, T., ... & Eliason, E. J. (2020). Maxed out: optimizing accuracy, precision, and power for field measures of maximum metabolic rate in fishes. *Physiological and Biochemical Zoology*, 93(3), 243-254.

Zhang, Y., Gilbert, M. J., & Farrell, A. P. (2020). Measuring maximum oxygen uptake with an incremental swimming test and by chasing rainbow trout to exhaustion inside a respirometry chamber yields the same results. *Journal of Fish Biology*, 97(1), 28-38.

8) L 463: These studies allowed [O₂] to drop as low as 80% air saturation and in these types of studies [O₂] rarely reaches 100% air saturation during flushing. The statement that 'SMR and MMR were measured at 21kPa' is thus not accurate. I would assume whatever corrective action or sensitivity analysis would be done to address this would show a similar pattern of results, but it should be addressed regardless.

9) L466: This is as not as universally accepted as stated here. I would state that "we assumed O₂crit of MMR to be 21kPa..."

Reviewer #2

(Remarks to the Author)

This study cleverly links aerobic scope with ecological fitness in wild Chinook salmon in California by calculating how metabolic index relates to fry rearing and smolt migration. The research integrated laboratory experiments, fine-scale field tracking, and predator studies to link physiology with management-relevant outcomes.

I commend the authors for the creativity they employed in this study. I think it's great for the field of ecophysiology that we're seeing more studies embracing meta-analysis and data synthesis of physiological data. In general, I quite enjoyed reading the manuscript.

I was asked specifically to gauge the author's rebuttal to reviewer 1's comments. Overall, I found Reviewer 1's criticisms valid and constructive. I also found the author's responses corresponding edits appropriate and satisfactory. I could not find an instance where I found the author's response unsatisfactory. I would suggest that the revised manuscript is appropriate for publication in its current form.

Version 2:

Reviewer comments:

Reviewer #1

(Remarks to the Author)

The authors have adequately addressed concerns raised in my previous review and now acknowledge limitations/assumptions in the data used to generate their models. More could be done to test these limitations and assumptions but given the substantial nature of the manuscript and analysis as is I do not think it is needed for publication.

Dear Dr. Repetto,

Thank you for the opportunity to revise our manuscript (COMMSBIO-24-8737-T) for consideration at *Communications Biology*.

In this revision, we have carefully considered the feedback provided by the three Reviewers. All author responses are colored in blue and line numbers reference the “clean” version of the revised manuscript without track changes. A version of the manuscript with track changes highlighted is also included with this revision.

In response to comments by all three Reviewers, we re-arranged the text to have a combined Results and Discussion followed by a brief Conclusions. In response to comments by Reviewers 1 and 2, we have re-arranged and added some paragraphs to the (now) Results and Discussion to clarify our findings, connect them to potential management options, and highlight some limitations that could be addressed by future work. In response to a comment by Reviewer 1, we have added model equations to the supporting information and now reference these in the main text where relevant.

While additional smaller edits were made to the text, figures, and tables to accomplish the improved clarity and transparency requested by the Reviewers, no methods or results have been changed in this revision.

Please note that, per our email discussion, we have begun our responses to Reviewer 1’s comments at their “overall comments” section to avoid redundancy.

Thank you again for the opportunity to revise our manuscript. We look forward to hearing from you soon.

Sincerely,
The Authors

Reviewer #1

OVERALL COMMENTS

The authors examined in Chinook salmon in the Sacramento-San Joaquin Delta two ecological fitness traits (fry/subyearling rearing and smolt/yearling migration) based on estimates of their metabolic index (ϕ) derived from their aerobic scope.

The authors determined that not only is the critical threshold of ϕ (ϕ_{crit}) important, but that a stable metabolic index (ϕ_{stable}), in which increases of aerobic scope do not yield benefits to fitness, is important to determine too. The authors claim that opportunities for management to recover these endangered Chinook salmon lie between ϕ_{crit} and ϕ_{stable} .

This conclusion arises from their study, which combines field research with tagged Chinook salmon at a refined spatial and temporal scale, laboratory experiments, and field research with a salmon predator.

They found that smolts had greater temperature sensitivity of hypoxia tolerance (based on parameter $-E$) than fry, and that they had higher hypoxia tolerance (based on parameter A) than fry. In comparing ϕ_{crit} to ϕ_{stable} , determined for fry and smolts separately, the authors found opportunities that would benefit the two ecological fitness traits of rearing and successful migration, but that these occurred in narrow range of environmental conditions. Through the year and seasons, the authors found that ϕ was high in the winter, when oxygen available is high in the environment and the demand from the organism is low, and vice versa the summer; and that these patterns were more pronounced in smolts.

With results from laboratory experiments, the authors found that their estimates of ϕ_{stable} fell within the estimates of ϕ of a fundamental niche. The authors interpreted the large differences between the upper temperatures associated with the fundamental niche and the temperatures associated with ϕ_{stable} for fry and for smolts as potentially a need for captively reared fishes to undergo rapid phenotypic plasticity to exploit the full realized niche in the wild.

The authors found that largemouth bass, a non-native predator of juvenile salmon, was more hypoxia-tolerant and had greater temperature sensitivity of hypoxia tolerance than salmon fry and smolts. The largemouth bass always had an aerobic scope surplus that was greater than that of juvenile Chinook salmon. Through the seasons, the ϕ associated with predation probability showed an opposite pattern than the ϕ associated with ecological fitness traits of juvenile salmon. The authors interpret these results as predation likely contributing to the Chinook salmon fitness detriments. Predation did not occur when there was the highest aerobic scoped advantage during cold temperatures, but predation probability was highest when the aerobic scope over prey was small and below ϕ_{stable} .

As a whole, the authors conducted a research study that was creative, and that covered many different aspects of ecology and physiology, with specific details aimed at being usable for management. The authors clearly put a lot of work and effort into this research study that is really multiple studies in one.

We find the Reviewer's summary of the paper to be accurate, and appreciate their compliments. This Reviewer's thorough feedback is greatly appreciated, and we have diligently worked to incorporate their suggestions in this revision.

However, I question the extent to which the main study findings related to ϕ_{stable} would truly be usable because of the representativeness of the ecological fitness traits the authors studied. I interpret fry rearing as essentially being the quantity of habitat available for fry. There would be major assumptions about how fry survival and abundance would be related to the available rearing habitat. For smolt migration, the authors did mention that this relates to successful migration and thus there is a component of survival within this migration fitness trait.

The authors relied on the last detection site which had nearly 100% detection. With regards to the temperatures the juveniles encountered at sites throughout the Delta above this last detection site would be important for accurate association to what the juveniles experienced through the Delta. There are some important assumptions here too because smolts can migrate through the delta and not be detected through various routes. A more appropriate model would be a mark-recapture model that provides probability estimates of detection and of survival. If the authors could provide sufficient and justifiable explanations for their choices in their methods with more detail or if the authors could incorporate probabilities of survival determined from mark-recapture models, then their study conclusions would be more robust. There are many studies of Chinook salmon in the Delta that have determined reach survival estimates and probabilities of detection using mark-recapture models (e.g., Buchanan et al. 2018 N. Am. J. Fish. Manag.; Perry et al. 2018 Can. J. Fish. Aquat. Sci.; Hance et al. 2021 Can. J. Fish. Aquat. Sci.).

We thank the Reviewer for the valid critiques, and have edited and made additions to the text to address them.

First, we have explicitly acknowledged the fry rearing assumption and also supported this assumption with a relevant reference by editing a sentence and adding a new sentence in the (now) Results and Discussion (lines 188-193): “Ecological fitness encompasses the ability of individuals to grow and survive (9); we therefore examined both the probability that rearing wild fry occupied potential Delta habitat, and the probability that captivity-raised smolts survived through-Delta migration. Given that the former was not a direct measure of growth or survival, we assumed that the availability of suitable rearing habitat was related to ability of fry to grow and survive, which has been shown in other regions of this watershed (47).”

We have also added lines 361-370: “Given the constraints of available data, two key limitations of our study must be acknowledged. First, our metric of fry rearing did not directly measure growth or survival, but assumed that both were related to the availability of suitable rearing habitat (e.g., 47). Second, environmental data were synchronized with smolt migration attempts post-hoc using water quality gauges, and we used average values across the duration of migration as independent variables (e.g., 30). For the former, future research might use advances in the miniaturization of acoustic tags to explore how the growth and survival of fry is related to habitat suitability on an individual basis. For the latter, future research might use temperature-logging acoustic tags or explore different methods of summarizing environmental conditions during migration (e.g., minima or duration at minima instead of averages).”

Second, in the Methods we have now clarified how we assigned smolts to Delta migratory routes, acknowledged the relevant assumptions made, and justified our strategy through edits and additions to the text (lines 530-540): “Using timeseries of detections for each tagged smolt, as revealed by acoustic receivers positioned throughout the Delta (Figure 2A, Figure S10), we determined that three general routes were taken during the study period: the Yolo Bypass, Sacramento River, and Interior Delta (Figure S10). We assumed that detections accurately represented smolt routing, and therefore assigned each smolt to the route it was detected within. However, if detections were missed (i.e., a smolt took a given route but was not

detected within the route), this could have biased route assignment and therefore the environmental parameters associated with the migration attempt. Given that there was greater temporal vs. spatial variation of ϕ in the Delta (Figure 4A, Figure S2), the impact of route assignment on the ϕ associated with each migration attempt was likely less than the timing of the migration attempt, which was quantified using receivers with near-perfect detection efficiency.”

Third, in response to comments by Reviewer #2 (but it also applicable here), we have acknowledged 3 assumptions and potential caveats relevant to how we associated environmental conditions to migrating smolts in the Methods through edits and additions to the text.

1. Lines 547-551: “Given that the resolution of both the environmental and acoustic telemetry data were similar, we assumed that environmental conditions as recorded by USGS stations were adequately representative of conditions experienced by migrating smolts. However, it is possible that finer-scale movements could reveal that migrating smolts are experiencing conditions that are distinct from those recorded by USGS stations.”
2. Lines 552-557: “Each smolt was associated with the mean value of route-specific temperature and O₂ partial pressure for the time it was present, presumed to be present (if missing detections near upstream end), or would have been present (if missing detections near downstream end) in the Delta. Note that Yolo Bypass and Interior Delta routes eventually rejoin the Sacramento River (Figure S10). While averaged environmental conditions were used in this study, it is important to note that other summarizing methods (e.g., maxima or minima) could be relevant.” Also, see lines 361-370 (text included earlier in this response).
3. Lines 557-562: “Missing detections at the upstream and/or downstream end of the Delta were interpolated using the shortest river distance between the temporally closest detection and the missing location using the “riverdist” package (83), and the most precise average travel rate possible (release group by route, all fish by route, or overall average for all routes and fish). We made the necessary assumption that the travel rate of other smolts was representative of the travel rate of smolts with missing detections.”

Finally, we have added an explanation and justification for why we chose GAMs to model smolt migration instead of mark-recapture models in the Methods (lines 564-570): “Similar to a previous telemetry analysis that leveraged near-perfect detection efficiency (30), generalized additive models (GAMs) (80) were used to assess predictor effects on through-Delta migration success probability of smolts. Compared to traditional mark-recapture models (which do incorporate detection efficiency), GAMs have greater flexibility for non-parametric predictor-response associations and more comprehensive residual diagnostics. This not only allowed us to examine hypothesized nonlinearities, but also to ensure that all model assumptions were met (e.g., Figure S11, Figure S12).”

Another reservation I have about the applicability of the findings is the extent to which managers can apply recommendations stemming from the study findings, given the amount of

cool water available from the Shasta Reservoir. The amount of water released from Shasta Reservoir would depend on the temperature and amount of water available and how much would be needed to cool temperatures in the Delta. The distance between Shasta Dam and the Delta is quite long and the water released from the reservoir would be warming up as it travels downstream to the Delta. In years when it would be necessary to cool temperatures in the Delta, there would also be likely a shortage of cool water available in the reservoir. If the authors could provide more discussion on the applicability of their findings for management. Especially with the narrow subset of viable environmental conditions that the authors determined attributable to fitness benefits, a discussion that connects findings to management options would be helpful to dissuade any doubts or uncertainty that would prevent any considerations for application.

The Reviewer brings up valid points, but it was a conscious decision on our part to avoid recommending specific management actions in the manuscript because we felt that would be outside the scope of this paper and better suited to the skillset of practitioners. Rather, one goal of the paper was to provide a mechanistic basis upon which potential management actions determined by others could rest. We therefore have taken this opportunity to flesh out when and why ϕ is a useful metric for managers to consider. This involved adding two new paragraphs and moving two existing paragraphs that were previously elsewhere in the text to make a new section in the (now) Results and Discussion called “Applicability of aerobic scope-based management actions in the study system” (lines 304-379). In this section, we succinctly discuss when and why ϕ can be a more relevant metric of water quality than temperature, connect our findings to potential management options, and highlight some limitations that could be addressed by future work.

In response to a later comment by this Reviewer, we have added a new sentence to the Introduction that clarifies that previous studies have quantified how reservoir releases affect downstream temperature in this watershed (lines 108-110): “Furthermore, multiple studies have shown how reservoir releases could be used within this watershed to regulate downstream temperatures (e.g., 33, 29).” This sentiment has also been added to the (now) Results and Discussion (lines 371-374): “While the impact of reservoir releases on downstream temperatures are becoming increasingly resolved in the study system (33, 29), AS-based management actions in the Delta are additionally limited by a dearth of information on how dissolved O₂ could be increased over short and long timescales.”

We have also added the sentiment that cold water availability need not restrict managing to moderate AS-related fitness detriments in the (now) Results and Discussion (lines 356-360, the second sentence is the added one): “Temperature regulation could have the added benefit of limiting mortality due to predation, either by reducing predator activity (40, 42), or by enhancing prey aerobic recovery between anerobic bouts of predator escape (35). However, even if cold water supply is limited, our models suggest that any increase in flow could offset AS-related fitness detriments when $\phi \leq \phi_{\text{stable}}$ (Figure 2D, E).”

On a related note, due to how the system is managed, even in wet years water temps can be excessively warm for juvenile salmon in the Delta when smolts migrate. A good example is the

current water year (2025) (using the following link, navigate to section 1.5):
https://oceanview.pfeg.noaa.gov/CalFishTrack/pageSEASON_2025.html

Finally, we have made some relevant adjustments to Figure 5 and its caption to help with clarity for managers. First, we changed the order of the axes so that temperature is now on the X and oxygen on the Y, which is how many temperature-related figures are presented in studies of this system. Second, we reiterated some key information at the beginning of the caption (lines 710-713): “ ϕ quantifies the synergistic impact of temperature and O_2 on aerobic scope (ϕ_f =fry-specific ϕ , and ϕ_s =smolt-specific ϕ). Flow does not change ecological thresholds of ϕ (i.e., ϕ_{crit} and ϕ_{stable}), only the baseline probability of fry rearing habitat use and smolt migration success (Figure 2D, E).” Third, we altered the figure and end of the caption to flesh out potential management scenarios, and the general options that our models suggest would be relevant (lines 719-732): “Circled numbers represent potential temperature and O_2 scenarios corresponding to the following general guidance: 1) conditions are above ϕ_{stable} and predation is below average—temperature reductions and O_2 increases would have little impact on rearing or migration but increases in flow might be beneficial; 2) conditions are between ϕ_{crit} and ϕ_{stable} and predation is above average—relatively small temperature reductions and O_2 increases would result in relatively large benefits to migration and rearing, as would increases in flow, in part by mitigating predation threat; 3) conditions are below ϕ_{crit} and predation is below average—temperature reductions and O_2 increases might be beneficial for rearing and migration (as might increases in flow), but alterations would likely have to be relatively large to see any benefit. Note that: ϕ_{crit} and ϕ_{stable} contours for fry rearing change little with temperature but drastically with O_2 , and ϕ_{crit} and ϕ_{stable} contours for smolt migration change similarly with both temperature and O_2 . Thus, when appropriate, temperature reductions and O_2 increases are both viable strategies to improve smolt migration success, while O_2 increases would be most impactful for fry rearing.”

With the results as currently analyzed and the study as written, the conclusions and claims are weakly supported. Please see comments about the representativeness of the ecological fitness traits, the need for mark-recapture modeling, and applicability to management application. Additional analysis, interpretation of results, and discussions of limitations would be needed for a more robust study.

Thank you for the feedback – please see our responses to the previous and subsequent comments, which explain the changes we have made to better balance these critiques with the strengths/robustness of our study.

MAJOR COMMENTS

Title

The title could be revised for a more direct account of what the study investigated and found. The key opportunities to recover imperiled salmon populations seems to be a stretch. Juvenile rearing habitat and successful migration would be more accurate than the term fitness.

We appreciate the Reviewer's feedback on our title, and have removed the word "key" and added the words "potential" and "help". It now reads: "Linking aerobic scope to fitness in the wild reveals potential opportunities to help recover imperiled salmon populations." If the editor would like further changes, please let us know.

Abstract

The abstract was well written to reflect the study as currently written.

Thanks. Some slight changes have been made to the Abstract in response to comments by Reviewers 2 and 3.

Introduction

The introduction is overall well written. The introduction could include more background on the different populations of Chinook salmon and when the juveniles migrate in through the Delta. A description of when the winter-run, spring-run, fall-run, and late-fall-run rear and migrate through the Delta would provide readers a better sense of what range of temperatures each population may encounter and when management may be most interested in the applicability of study findings.

We are glad to hear the Reviewer found the Introduction well-written. We have now included the requested information on the phenology of fry and smolts of different populations in the Delta (lines 94-99): "While populations are named for the season when adults return to freshwater from the ocean, the timing of fry rearing in, and smolt migration through, the Delta is also somewhat distinct. During fall and winter, fry of winter-returning, spring-returning, and fall-returning adults are usually present, as well as smolts of winter-returning adults. During spring and summer, fry of late fall-returning adults and smolts of spring-returning, fall-returning, and late fall-returning adults tend to be present (29, 30)."

In lines 111-112, I wondered about how much cool water release would be needed from the Shasta Reservoir to cool the Delta. I think some more details and specificity about this system is needed in the introduction for readers to ease any doubts they may have about applicability. As previously mentioned, we have added the following sentence to the end of this paragraph (lines 108-110): "Furthermore, multiple studies have shown how reservoir releases could be used within this watershed to regulate downstream temperatures (e.g., 33, 29)." This sentiment has also been added to the (now) Results and Discussion (lines 371-374): "While the impact of reservoir releases on downstream temperatures are becoming increasingly resolved in the study system (33, 29), AS-based management actions in the Delta are additionally limited by a dearth of information on how dissolved O₂ could be increased over short and long timescales."

Results

Overall, the topics within each paragraph is logical and generally flows well through the manuscript. They were:

1. Methods (equation 1) related to main study metric (metabolic index, phi)

2. Interpretation of parameters of equation 1
3. Parameter -E
4. Parameter A

5. phi
6. phi_crit
7. phi_stable
8. flow
9. fundamental niche
10. limitation of temperature explanation for phi (This paragraph in lines 212-221 was a bit confusing and could be edited for clarity.)

11. Seasonal cycle
12. Spatiotemporally averaged phi
13. Predator, largemouth bass
14. Predator and prey phi comparisons
15. Burst escape, aerobic debt
16. Conclusion

We are glad to hear the Reviewer found the flow of topics in the Results (now Results and Discussion) logical. We have edited the limitations of temperature paragraph (# 10 in the Reviewer's list) for clarity and moved its position to help address another comment by this Reviewer. Please see lines 305-322 for the revised version (now the first paragraph of the new section "Applicability of aerobic scope-based management actions in the study system").

Given the amount of content, I did find myself lost and an introductory paragraph for the Results section that lists the upcoming subsections would be helpful. The last paragraph of the introduction could be another place where the authors could write out the objectives so that readers know what to expect in the Results section.

We apologize for the confusion. We had the list of objectives and broad overview of methods in the second to last paragraph of the Introduction. We have now swapped the position of these two paragraphs, which should improve clarity. See lines 100-122.

The remark in lines 133-134 about differences between populations and lifestages is appreciated.

We are glad to hear this.

The linear mixed effects regression that the authors applied was with populations being the random effect. However, there were only 4 populations. Usually, the rule of thumb is having at least 7 observations or groups for a random effects distribution. Four would be insufficient and the authors probably could explain why they think population is justifiable for a random effect rather than fixed effects. I see each of these four runs of salmon being quite distinct in their life history traits. I, and perhaps other readers, would also be curious to know how much random effects explained the variation in the data. In line 139, does the R^2 include or exclude random effects?

We agree with the Reviewer, which is why we presented results from two models side-by-side: one where population was a fixed effect (the MR), and one where population was a mixed effect (the LMER) (please see Figure 1 and Table 1).

We have added a sentence at the end of this paragraph justifying and clarifying why we used the LMER parameters in the rest of the manuscript (lines 158-160): “Given the congruence of MR and LMER predictions (Figure 1), we used metabolic traits averaged across populations (i.e., parameters from the LMER in Table 1) to produce ϕ for each lifestage that was broadly applicable across populations in the study system.”

Great suggestion – we have now specified the conditional (fixed and random effects) vs. marginal (fixed effects only) R^2 for the LMER (lines 147-151): “Temperature sensitivities ($\pm 95\%$ confidence intervals, CIs) for fry ($-E=0.16\pm 0.06$) and smolts averaged across populations ($-E=0.33\pm 0.03$) (linear mixed-effects regression [LMER]: conditional $R^2=0.54$, marginal $R^2=0.48$, residual $df=634$) (Table 1, Equation S1) fell in the 20% and 50% quantiles, respectively, of a study of the thermal sensitivities of diverse ectothermic taxa (5).”

For the effect from body size (line 149), does this differ between Chinook salmon runs? Interesting question. Unfortunately, we did not have the data resolution to test this (i.e., for some populations there was not a sufficient size range). However, we suspect that any difference between populations would be minor given the minimal population-specific variation in metabolic traits. We have now noted that we did not test for population-specific mass scaling in the Methods (lines 474-476): “We did not test for population-specific mass scaling because not all populations were represented with a sufficient size range in the data.”

Regarding the ontogenetic variation (line 154...), I was confused about how fry tend to rear in warmer reaches and how they are less hypoxia tolerant. Could the authors please explain this further? Furthermore, the text about “...while smolts migrate to a cold marine ecosystem” (line 156), it would seem more appropriate to write about how smolts may seek cooler temperatures during their downstream migration.

Ok – we have added two sentences and updated a third in this section for clarity (lines 167-175): “In other words, the O_2 demand of SMR for fry is relatively insensitive to temperature, allowing this lifestage to inhabit warmer or cooler waters with minimal impact to AS. The SMR of smolts, on the other hand, requires relatively higher O_2 when warmer (restricting AS), but relatively less O_2 when cooler (expanding AS). This ontogenetic variation likely reflects physiological adaptations to distinct environmental or energetic challenges (21): fry tend to move from cooler upper river reaches to rear in warmer downstream reaches (29), while smolts seek cooler temperatures during downstream migration, the destination of which is a cold marine ecosystem with shallow layers of hypoxia (3, 6, 22).”

When the authors write about exceptionally high flows (lines 197-198), can the authors please provide more information about whether this is relevant in most years or only in wet water years? What about water years that are dry or critically dry, when managing for cooler temperatures would be most important?

This is a good point. We have now clarified when flow vs. ϕ could be used in a compensatory manner by adding a detail to one of the sentences in this paragraph (lines 225-226): “Habitat quality and quantity can therefore offset one another in order to achieve the same fitness benefit when $\phi \leq \phi_{\text{stable}}$.”

In addition, we have added a sentence to a later paragraph in the (now) Results and Discussion (lines 358-360): “However, even if cold water supply is limited, our models suggest that any increase in flow could offset AS-related fitness detriments when $\phi \leq \phi_{\text{stable}}$ (Figure 2D, E).”

In various places throughout the manuscript, the authors could be clearer about “while averaging over other parameters” and a reference to an equation. There were many analyses that included linear regressions with and without random effects, GAMs with and without random effects, etc. and including numbered equations for these in the Suppl. Mat would help readers easily and quickly track what the authors did. Otherwise, it is difficult and time-consuming to draw out from the text.

Thank you for the suggestion – equations are now included in the Supplementary Information and referenced in the main text.

Discussion

The discussion was relatively short and many portions in the results section may be better placed in the discussion. The Results section was 16 paragraphs long, while the discussion was only 3 paragraphs long. In other articles in Nature Communication, it seems like the discussion section is more on the order of about 9-10 paragraphs, but can range as high as 13 paragraphs, and, in one case that I saw, as low as 2 paragraphs.

In light of this helpful comment, we felt that it was most space efficient and best aided comprehension for readers, to include contextualizing discussion alongside the relevant results in this revision. To do this, we have changed the title of the “Results” section to “Results and Discussion” and re-labeled the “Discussion” section to “Conclusions.”

There were several places where the authors mention very generally “other important environmental factors”. If the authors could expand even a little with the mention of which these environmental factors are, this would help provide readers with a better understanding of which factors the authors are referring to and the extent to which managers could gage study findings in a broader but specific context.

Thank you for the suggestion – we have added more detail in two key instances, both in the second section of the Results (now Results and Discussion). Lines 184-188: “Using ϕ specific to Chinook salmon fry and smolt lifestages averaged across populations (LMER parameters in Table 1), we assessed if and how AS, alongside other key environmental factors such as flow (volumetric discharge of the Sacramento River, the primary watershed feeding the Delta), was related to ecological fitness in the study system (Figure 2A).” Lines 209-211: “Beyond ϕ_{stable} , additional ϕ was not necessarily advantageous, and its influence on fitness likely waned compared to other environmental factors (e.g., flow).” For a third instance (lines 281-284), we

deleted “other environmental factors” because we realized that other factors in the model weren’t environmental per se, but more related to the study method.

In addition, now that equations are included in the Supplementary Information and referenced throughout the main text (see our response to a previous comment by this Reviewer), it should be easier for a reader to quickly lookup further details on the models if they so choose.

I appreciate the authors interpretation of results with three types of hypotheses that respectively follow a limiting, facilitating, and circumstantially important interpretations of findings under a natural setting. However, the authors then don’t provide a clear answer within this structure of three hypotheses. It seems that the authors could present the conclusions in a better way than setting this structure only to immediately not follow it.

We are glad to hear the Reviewer appreciated the structure of the three hypotheses, and agree that we could have more clearly presented a relevant conclusion.

In the relevant section of the (now) Results and Discussion, we have changed the wording and position of a key sentence so that it is clearer and appears at the beginning of this section (lines 183-184): “Our results suggest that AS can circumstantially both limit and facilitate the ecological fitness of Chinook salmon in the wild.”

We have also rearranged the relevant part of the (now) Conclusions. Please see the second paragraph of this section (lines 391-409). This paragraph was originally two separate paragraphs, which we combined to help with flow and clarity.

Further discussion on other temperature-related detriments to fitness, such as disease, physiological development, and food resources, could be included. It seemed that the study was quite focused on aerobic scope, and I (and perhaps other readers) would wonder about other temperature-related factors.

Good point. As specified in a previous response to this Reviewer, we have made a new section in the (now) Results and Discussion called “Applicability of aerobic scope-based management actions in the study system” (lines 304-379). In this section, we succinctly discuss when and why ϕ can be a more relevant metric of water quality than temperature, connect our findings to potential management options, and highlight some limitations that could be addressed by future work.

The discussion could also expand on the realism of application of study findings for management. It could include a discussion on: how much cool water is expected to be available in particular water year types, how much water would be needed for release to cool temperatures in the Delta, and whether there is indeed opportunity for management to improve on decisions based on the study findings related to ϕ_{stable} . Managers may feel like they can only manage to ϕ_{crit} , given the amount of water available, the uncertainty in findings, and the narrow range that’s needed to be managed to.

A detailed discussion on water availability in different water year types could be interesting, but we feel that this is outside the scope of our study. We have, however, referenced several

detailed hydrology studies where interested readers could learn more about the state-of-the-art of water management for downstream temperature in this system.

Methods

L428. Why was an interaction between flow and phi included? The choice of including (or not including) an interaction factor should be explained beforehand and not rely just rely on whether there was a significant effect or not.

Ok – we have elaborated on why an interaction was included.

For the rearing GAMMs (lines 513-515): “To explore a hypothesized synergistic impact of water quantity and quality on rearing habitat use, a parametric interaction between the flow and ϕ or temperature smooth terms was additionally included.” And for the migration GAMs (lines 574-578): “As with the rearing analysis, we investigated a hypothesized synergistic impact of water quantity and quality on migration success using interactions between flow and ϕ . A parametric interaction was included in the ϕ GAM only, as the corresponding interaction was not supported in the temperature GAM given that it masked the effect of flow.”

L447-448. How often were temperature and O2 missing?

We have now added this information (lines 543-544): “Periods of time where temperature or O₂ data were missing were excluded from analysis (7% excluded).”

L451-452. Can the authors provide further detail or assumptions on how the temperature and O2 partial pressure data were associated with smolt passage routes or detections?

We have now provided more details and assumptions.

First, we have edited and made additions to the paragraph that explained how we determined routing. It now reads (lines 530-536): “Using timeseries of detections for each tagged smolt, as revealed by acoustic receivers positioned throughout the Delta (Figure 2A, Figure S10), we determined that three general routes were taken during the study period: the Yolo Bypass, Sacramento River, and Interior Delta (Figure S10). We assumed that detections accurately represented smolt routing, and therefore assigned each smolt to the route it was detected within. However, if detections were missed (i.e., a smolt took a given route but was not detected within the route), this could have biased route assignment and therefore the environmental parameters associated with the migration attempt.”

Next, we added two sentences to the end of the paragraph where temperature and O₂ data sources were described (lines 546-551): “Given that the resolution of both the environmental and acoustic telemetry data were similar, we assumed that environmental conditions as recorded by USGS stations were adequately representative of conditions experienced by migrating smolts. However, it is possible that finer-scale movements could reveal that migrating smolts are experiencing conditions that are distinct from those recorded by USGS stations.”

Finally, we added two sentences to the paragraph where the process of associating smolt migrations and environmental conditions was described (lines 556-562) (the added sentences are the first and last of the following): “While averaged environmental conditions were used in this study, it is important to note that other summarizing methods (e.g., maxima or minima) could be relevant. Missing detections at the upstream and/or downstream end of the Delta were interpolated using the shortest river distance between the temporally closest detection and the missing location using the “riverdist” package (83), and the most precise average travel rate possible (release group by route, all fish by route, or overall average for all routes and fish). We made the necessary assumption that the travel rate of other smolts was representative of the travel rate of smolts with missing detections. Mean tidally filtered flow was also associated with each smolt.”

The authors seem to be using a mix of maximum likelihood and Bayesian methods for different analyses. Can the authors please explain and justify their choices? It seems strange for a single study to use a mix and not set out to run analyses all in one or the other. The mix of terms and concepts like bootstrapping and BIC across different analyses seems strange.

We are more than happy to explain. The only instance where “Bayesian” is reported in the manuscript is in reference to Bayesian information criterion (BIC). However, this name is somewhat confusing given that BIC is actually based on maximum likelihood estimation. The formula used for BIC is:

$$-2 \ln(\text{likelihood}) + k \ln(N)$$

where *likelihood* is obtained from a model object fitted using maximum likelihood (as was done for all models in our study), *k* is the penalty per parameter (the default is 2), and *N* is the number of observations. We used BIC for all model comparison exercises because its use is recommended when the goal is explanation rather than prediction.

Re: bootstrapping, we realized that it was included in Figure 2’s caption in error. We have therefore deleted this occurrence (lines 665-667): “Light lines and grey ribbons respectively show predictions and 95% CIs for how probabilities changed across the full range of sampled ϕ , while averaging over other parameters (see Equation S3, Equation S4).”

Therefore, the only part of our study where bootstrapping was used was to generate confidence intervals around estimates from the linear mixed effects regression (LMER) where we determined metabolic traits for fry and smolt lifestages averaged across populations. We used the “lme4” package in R to fit this LMER. The authors of this package recommend bootstrapping to generating confidence intervals for LMERS because bootstrapping can incorporate variance around the random effect, leading to more accurate (usually wider) confidence intervals.

We have clarified our use of bootstrapping in the Methods (lines 480-484): “To determine lifestage-specific metabolic traits averaged across populations, we fit a linear mixed-effects regression (LMER) with a by-population random intercept (Equation S1) using the “lme4” package (78). The “bootMer” function in this package was used to generate bootstrapped confidence intervals for LMER predictions that incorporated variance around the random

intercept (Figure 1).”

Figures

Figure 1. Inverse kinetic energy was first mentioned and described in Methods. I was confused as I was pretty sure I had not encountered this term yet while reading the Introduction and the Results. It would probably help readers if the authors could explain this term in the text of the results so that readers don't encounter it for the first time in Figure 1. Furthermore, it might be easier to track the second x-axis (for temperature) below the first x-axis (inverse kinetic energy), rather than above the panel titles (fry and smolt).

We appreciate the Reviewer drawing our attention the lack of explanation for inverse kinetic energy. We realized that “inverse temperature” is probably a more appropriate term for the audience of our paper, and have updated all appearances of “inverse kinetic energy” in the main text and supplementary information accordingly.

Furthermore, we have better explained why inverse temperature is used when parameterizing the metabolic index in Figure 1's caption (lines 645-651): “The intercept and slope of the associations between hypoxia tolerance and inverse temperature supplied the metabolic traits A and $-E$, respectively, for each lifestage (Table 1). These traits were then used to parameterize the metabolic index (Equation 1) so that environmental temperature and O_2 could be converted into ϕ (i.e., FAS). For the units to cancel out in this equation, it is necessary for temperature to take inverse format in eV; however, for the convenience of the reader, the secondary x-axis (above) shows the corresponding temperature in $^{\circ}C$.”

While we appreciate the suggestion by the Reviewer to move the secondary x-axis to the bottom of the plot (below inverse temperature), we could not find an efficient route to do so in the R package used for plotting (ggplot2). We feel that the changes to the x-axis label and increased clarity in the figure caption make the figure easier to understand.

Figure 2. It took me a while to wrap my head around “fitness benefit” on the y-axes. I suggest the authors just state what these are for fry and for smolts. I understand that the authors may want to have a common term or metric for both juvenile life stages, but it seems more confusing than helpful, and gives the impression that the authors are overstressing conclusions. It also took me a while to understand “The red and blue lines respectively show how the density (presented on a 0 to 0.5 scale) of 0 and 1 observations in each dataset aligned with ϕ .” I interpret it as something similar to the distribution of a run passage across the season, but this is of either the “0” observations (red line) or the “1” observations (blue line) across a gradient of ϕ . Perhaps the authors could edit for easier understanding for readers. Can the authors also provide an explanation in the methods for why 0.01-0.99 quantiles were chosen here while the rest of the paper use 95% for the CI? Why not use 0.025-0.975 quantiles? “Fitness benefit” on the y-axes has now been changed to “rearing habitat use” in Figure 2B, D and “migration success” in Figure 2C, E. Please see the new version of Figure 2.

We have also rearranged the sentence describing the red and blue lines for clarity (lines 667-669): “The red and blue lines respectively show how the density of “0” and “1” observations in each dataset aligned with ϕ , and are presented on a 0 to 0.5 scale.”

Finally, we edited the explanation of panels D and E in Figure 2 to clarify why we chose restrict predictions to the 0.01-0.99 quantiles for flow (lines 670-673): “(D & E) Gradients show predicted probabilities of (D) fry rearing habitat use and (E) smolt migration success under potential flow by ϕ combinations. Predictions are only shown across 0.01-0.99 flow quantiles due to the high uncertainty associated with rare flow conditions.”

The authors may want to consider listing years in all figure captions or not in a few of them. Thank you for the suggestion - we have added years to the first sentence of Figure 2’s caption. Years were already present in Figure 4’s caption, and we didn’t feel that other figures warranted the inclusion of years.

Figure 4. The y-axis label that has “+/- max uncertainty” does not match the 95% CI in the caption. Please edit for consistency.

Good catch, thank you. For the (A) panels in this plot, the lines are actually the 95% CIs of the model prediction, while the ribbons are the range of the raw data. We have updated the second sentence of the caption so that it now reads (lines 692-694): “Lines are the 95% CIs of spatiotemporal ϕ GAMs predicted at an average location within the study system (see Equation S5, Equation S6, and Equation S8).”

MINOR COMMENTS

L130. Instead of “larger -E”, I suggest editing to “more negative -E” to be more accurate.
Ok – we have changed this.

L187. Correct the typo “rage” to “range”.
Good catch. Corrected.

L416. “Length-at-date criteria” is probably more appropriate than “Length-at-date criterion” because there are multiple criteria for each of the runs. Please provide a reference for this method.
We have changed this as the Reviewer suggested and added a reference.

L320, 324, 327. The “)” after each numbered hypothesis can probably be removed. It causes confusion that is unnecessary. Because these are not in a list, but across a few sentences, readers may look for the opening parentheses.
Ok – these have been removed.

L426. I wasn’t sure what “by-sampling” was meant by the authors. It could use a brief explanation at first mention.

We feel that the addition of model equations to the Supplementary Information, which we added in response to an earlier suggestion by this Reviewer, have helped clarify the terms in this model.

L571. Change “Lines are the fits +/- 95% CIs...” to “Lines are the model fits and the shadings are the 95% CIs...” or something similar.

As specified in an earlier response, the lines in the (A) panels of this figure are actually the 95% CIs of the model prediction, and we have updated the second sentence of the caption accordingly.

Throughout the manuscript, the authors used “+/- 95%CI” or upper and lower 95%CI but the authors should change CI to confidence limits (CL) to be accurate. Alternatively, the authors could change it to “+/- 1.96 SD” or something along those lines.

Confidence limits are another name for the numbers at the end of the confidence intervals. Aside from the instance identified in the final comment by Reviewer 1, we found one other instance where confidence limit was more appropriate, the caption for Table 1. We have renamed accordingly in both instances, and changed the relevant column names in Table 1.

L593. Here is an example of “lower 95% CI” and “upper 95%” that should probably be edited to “lower 95% CL” and “upper 95% CL”.

Done.

Reviewer #2

The gap in the literature that the paper purports to address is whether observed correlations between aerobic scope and range limits comprise a fundamental limitation to species distributions or a circumstantial correlation that “reflects species tracking of preferred conditions for aerobic performance rather than barriers to their distributions”.

They claimed to have resolved this debate by showing a case in which AS is a fitness-limiting barrier (In 319-320), by which they meant that fry habitat use was better predicted by a phi + flow model than a temperature + flow model in a GAMM framework. I think the major evidence for this is Appendix Table S2, although there was no difference in performance of the two models for smolt survival. They also identified lots of situations that were easily within the physiological range, which they describe as facilitating or above phi stable. They also used predator experiments and phi calculations to show that warm water predators have interesting differences in their AS from salmon, and they are most effective at catching salmon when salmon are stressed.

We thank the Reviewer for their assessment of our paper, and recognize that this was a dense manuscript that, as Reviewer 1 put it, “is really multiple studies in one.” We made numerous changes for clarity and transparency in response to comments by Reviewer 1—in instances

where those changes are relevant for this Reviewer's comments, we have reiterated the changes here but not in an exhaustive manner.

We noticed a slight misunderstanding in this Reviewer's comment where they said, "fry habitat use was better predicted by a ϕ + flow model than a temperature + flow model in a GAMM framework." Accordingly, we have better emphasized the temperature vs. ϕ component of our results, and how this differed between fry and smolt lifestages. This involved editing a paragraph for clarity and moving it to the new section in the (now) Results and Discussion called "Applicability of aerobic scope-based management actions in the study system." Please see lines 305-322 for the edited version of this paragraph (the first in this new section).

In addition, where relevant, we have toned down the language used when discussing our results because we do not feel that our findings "have resolved this debate." The title has been changed from "Linking aerobic scope to fitness in the wild reveals key opportunities to recover imperiled salmon populations to" "Linking aerobic scope to fitness in the wild reveals potential opportunities to help recover imperiled salmon populations." Additionally, in the Abstract, we have replaced "governed" with "was associated with" in the following sentence (lines 9-12): "We found that AS, which we quantified using the metabolic index (ϕ), was associated with success probability for these bottlenecks only under a relatively narrow window of viable environmental conditions, depending on intraspecific metabolic trait diversity and hydrologic conditions." Also in the Abstract we have added "potentially" before "high-impact" and "could" before "therefore" in the following sentence (lines 12-14): "Opportunities for potentially high-impact temperature- and O_2 -specific conservation and management actions using existing hydraulic engineering infrastructure could therefore exist when AS is between critical (ϕ_{crit}) and stable (ϕ_{stable}) values."

Honestly, I had a hard time following the logic of the dominant argument. I believe we are still looking at a correlation between environmentally suitable conditions and fish behavior. I don't see how this changes range predictions with climate change or restoration priorities. It seems like what you were really trying to do is determine how quickly AS could evolve, because that would require understanding the most important selective pressure and any additional limiting factors. I don't see how just comparing performance does anything different from previous studies (albeit granted it is at much finer resolution, which is laudable). Even if it is "only circumstantially relevant for fitness in natural settings", it would still be correlated with fitness or habitat preference.

To help readers follow the dominant argument, we have clarified the broader takeaways of our study. In the relevant section of the (now) Results and Discussion, we have changed the wording and position of a key sentence so that it is clearer and appears at the beginning of this section (lines 183-184): "Our results suggest that AS can circumstantially both limit and facilitate the ecological fitness of Chinook salmon in the wild." We have also rearranged the relevant part of the (now) Conclusions (lines 391-409). This paragraph was originally two separate paragraphs, which we combined to help with flow and clarity.

The Reviewer is correct in that our work doesn't "[change] range predictions with climate change or restoration priorities." In reviewing the text, we couldn't figure out where they got the notion that this is what we were trying to accomplish—rather, we were testing the validity of a key assumption underlying these predictions. In an attempt to clarify this, we have deleted the last sentence of the Abstract so that it now ends with (lines 17-18): "In addition, AS impairments likely increased susceptibility to predation, and this may have been involved in the putative association between AS and fitness in the wild."

The resolution the Reviewer finds "laudable" is indeed what makes our study distinct and important because it allowed us to explicitly investigate potential associations between aerobic scope and ecological fitness. Prior work that had produced range predictions with climate change had to assume that aerobic scope was important for fitness—see our Introduction and (now) Conclusions. According to our results, the association between aerobic scope and fitness in the wild is actually quite nuanced and not as simple as assumed by the range projection studies. To clarify why this detail and nuance is important, we have added a paragraph to the (now) Results and Discussion that explores management implications. Please see lines 323-346, the second paragraph in the new section "Applicability of aerobic scope-based management actions in the study system."

I believe that O_2 is still strongly correlated with temperature and flow, so it would have been informative to show spatially how the restoration priority would be different using the alternative criterion, or other important distinctions. It probably does offer a more mechanistic explanation than we usually have. But flow has a lot of other impacts (e.g., it often determines the speed of migration, and therefore the duration of exposure to unhealthy conditions). I did not see that explicitly taken into account, except implicitly when averaging over the duration of the migration. But I feel like this averaging, given the strong seasonal patterns in flow and temperature, had a lot to do with the results. Perhaps there is a period of time over which exposure to hypoxia is critical, or some other way of evaluating their exposure during migration. O_2 is indeed correlated with temperature—by definition, ϕ accounts for the temperature-dependence of environmental O_2 and organismal O_2 demand (see Equation 1). Thus, our results encompass the synergistic impact of O_2 and temperature on performance. This reiteration has been added to the main text. Again, please see the newly added a paragraph to the (now) Results and Discussion that explores management implications (lines 323-346, the second paragraph in the new section "Applicability of aerobic scope-based management actions in the study system"). We have also reiterated the mechanistic nature of ϕ in the (now) Results and Discussion (lines 307-308): " ϕ , on the other hand, specifically quantifies the AS of a given habitat and therefore directly assesses the impact of temperature and O_2 on physiological performance." We have also reiterated this in the caption of Figure 5 (lines 710-712): " ϕ quantifies the synergistic impact of temperature and O_2 on aerobic scope (ϕ_f =fry-specific ϕ , and ϕ_s =smolt-specific ϕ)."

Any impact of flow was accounted for in our results because flow was considered alongside ϕ using an additive modeling framework. Accordingly, each independent variable explained distinct variation in the dependent variable, and this was verified with concurrency—thus, flow

and ϕ did not meaningfully covary. We have added a definition for concurrency to the Methods (lines 507-510): “Since ϕ and temperature were highly correlated, leading to excessive concurrency (≥ 0.7) (the extent to which independent variables approximate one another’s impact on the dependent variable, ranging from 0-1) when included in the same model, we assessed their effect using two separate models.”

Furthermore, in Response to a comment by Reviewer 1, we have better highlighted that we explored how flow and ϕ interacted (see Figure 2D, E). In the (now) Results and Discussion (lines 218-220): “Using interactions in the rearing GAMM and migration GAM, we investigated the how AS (a metric of water quality) and flow (a metric of water quantity) synergistically impacted fitness.” In the Methods: “To explore a hypothesized synergistic impact of water quantity and quality on rearing habitat use, a parametric interaction between the flow and ϕ or temperature smooth terms was additionally included” (lines 513-515) and “As with the rearing analysis, we investigated a hypothesized synergistic impact of water quantity and quality on migration success using interactions between flow and ϕ ” (lines 574-576).

We appreciate the Reviewer’s interest in spatial prioritization, which was the goal of Figure S2 and its caption. Notably, ϕ exhibited more temporal than spatial variation in the Delta, and we have added this sentiment to the main text in the Methods (lines 537-540): “Given that there was greater temporal vs. spatial variation of ϕ in the Delta (Figure 4A, Figure S2), the impact of route assignment on the ϕ associated with each migration attempt was likely less than the timing of the migration attempt, which was quantified using receivers with near-perfect detection efficiency.”

We agree that future efforts could examine in more granular detail when and how ϕ is circumstantially limiting or facilitating for fitness in the wild using our study system. As pointed out by the Reviewer, a promising avenue would be to explore where smolts encountered minimum ϕ and for how long. We added this sentiment to a new paragraph in the (now) Results and Discussion where we discuss key limitations to our study and recommend future work. Please see lines 361-379, the 4th and 5th paragraphs in the new section “Applicability of aerobic scope-based management actions in the study system.”

Finally, we have explicitly acknowledged 3 assumptions and potential caveats relevant to how we associated environmental conditions to migrating smolts in the Methods.

1. Lines 547-551: “Given that the resolution of both the environmental and acoustic telemetry data were similar, we assumed that environmental conditions as recorded by USGS stations were adequately representative of conditions experienced by migrating smolts. However, it is possible that finer-scale movements could reveal that migrating smolts are experiencing conditions that are distinct from those recorded by USGS stations.”
2. Lines 552-557: “Each smolt was associated with the mean value of route-specific temperature and O₂ partial pressure for the time it was present, presumed to be present (if missing detections near upstream end), or would have been present (if missing detections near downstream end) in the Delta. Note that Yolo Bypass and Interior Delta

routes eventually rejoin the Sacramento River (Figure S10). While averaged environmental conditions were used in this study, it is important to note that other summarizing methods (e.g., maxima or minima) could be relevant.” Also, see lines 361-370 (text included earlier in this response).

3. Lines 557-562: “Missing detections at the upstream and/or downstream end of the Delta were interpolated using the shortest river distance between the temporally closest detection and the missing location using the “riverdist” package (83), and the most precise average travel rate possible (release group by route, all fish by route, or overall average for all routes and fish). We made the necessary assumption that the travel rate of other smolts was representative of the travel rate of smolts with missing detections.”

From a practical standpoint, it is somewhat concerning that ϕ is so population- and life stage-specific. That sounds to me like it is not a fundamental limiting factor, but rather a secondary product reflecting prevailing conditions, and also not the most useful for management.

We thank the Reviewer for expressing their opinion, but respectfully disagree. Thermal sensitivities have long been recognized to vary between lifestages and populations of salmon, which (in some systems) has led to specific temperature thresholds being prescribed for management (e.g., the Columbia River basin, Richter & Kolmes 2005). The temperature sensitivity of O_2 demand was the fundamental limiting factor considered in our study, and ϕ is parameterized to reflect this fundamental limitation. Given differences in behavior and habitat between lifestages and populations of Chinook salmon in the Sacramento-San Joaquin watershed, that fundamental physiological limitations (and therefore ϕ) would also vary is unsurprising. As we explain in the manuscript, previous studies using ϕ have assumed that fundamental limitations do not vary within species—our work reveals this can be an incorrect assumption, which will help progress the field toward more accurate forecasts with climate change, restoration priorities, etc.

We agree that we could have better explained the utility of ϕ as an index for management in the study system. Thus, we have made some additions to the text to help clarify how and why ϕ could be used by managers. First and foremost is the “Applicability of aerobic scope-based management actions in the study system” section added to the (now) Results and Discussion (lines 304-379). In this section, we succinctly discuss when and why ϕ can be a more relevant metric of water quality than temperature, connect our findings to potential management options, and highlight some limitations that could be addressed by future work.

Also, we made some key updates to Figure 5 and its caption to help with clarity for managers. First, we changed the order of the axes so that temperature is now on the X and oxygen on the Y, which is how many temperature-related figures are presented in studies of this system. Second, we reiterated some key information at the beginning of the caption (lines 710-713): “ ϕ quantifies the synergistic impact of temperature and O_2 on aerobic scope (ϕ_f =fry-specific ϕ , and ϕ_s =smolt-specific ϕ). Flow does not change ecological thresholds of ϕ (i.e., ϕ_{crit} and ϕ_{stable}), only the baseline probability of fry rearing habitat use and smolt migration success (Figure 2D, E).” Third, we altered the figure and end of the caption to flesh out potential management scenarios, and the general options that our models suggest would be relevant (lines 719-732):

“Circled numbers represent potential temperature and O₂ scenarios corresponding to the following general guidance: 1) conditions are above ϕ_{stable} and predation is below average—temperature reductions and O₂ increases would have little impact on rearing or migration but increases in flow might be beneficial; 2) conditions are between ϕ_{crit} and ϕ_{stable} and predation is above average—relatively small temperature reductions and O₂ increases would result in relatively large benefits to migration and rearing, as would increases in flow, in part by mitigating predation threat; 3) conditions are below ϕ_{crit} and predation is below average—temperature reductions and O₂ increases might be beneficial for rearing and migration (as might increases in flow), but alterations would likely have to be relatively large to see any benefit. Note that: ϕ_{crit} and ϕ_{stable} contours for fry rearing change little with temperature but drastically with O₂, and ϕ_{crit} and ϕ_{stable} contours for smolt migration change similarly with both temperature and O₂. Thus, when appropriate, temperature reductions and O₂ increases are both viable strategies to improve smolt migration success, while O₂ increases would be most impactful for fry rearing.”

Most of the discussion of applications was in the Results section rather than the Discussion section. This placement would have made more sense if the authors showed something quantitative that compared the two conservation approaches. Maybe a map of the highest priority restoration sites under the two criteria (ie., using phi vs temp+flow, rather than just temp in Fig 5). Or maybe this metric provides a synthesizing variable that can bridge the temp/flow combination. It doesn't totally replace either one, so it is not a simpler regulatory system.

Reviewer 1 identified a similar issue. In this revision, we felt that it was most space efficient to include contextualizing discussion alongside the relevant results. To make this clearer, we have changed the title of the “Results” section to “Results and Discussion” and re-labeled the “Discussion” section to “Conclusions.”

As we have noted in previous responses, ϕ combines temperature and O₂ impacts on salmon into a single metric that can be used in conjunction with flow for management because their effects on both responses are additive. We have updated the caption of Figure 5 to reiterate this for clarity (lines 710-713): “ ϕ quantifies the synergistic impact of temperature and O₂ on aerobic scope (ϕ_f =fry-specific ϕ , and ϕ_s =smolt-specific ϕ). Flow does not change ecological thresholds of ϕ (i.e., ϕ_{crit} and ϕ_{stable}), only the baseline probability of fry rearing habitat use and smolt migration success (Figure 2D, E).”

Again, we appreciate the Reviewer's interest in spatial restoration priorities. As explained in a previous response, Figure S2 and its caption are relevant, but so is our finding that there is more temporal variation in ϕ than spatial variation in the study system. In the end, more detailed assessments of potential restoration sites would require finer-scale temperature and O₂ data than currently available. We have added this sentiment to the (now) Results and Discussion (lines 375-377): “While our study provides some guidance on where AS might benefit from habitat restoration (Figure S2), more detailed and systematic surveys of temperature and O₂ in the Delta would greatly assist in such efforts.”

The authors do present an interesting idea and have employed a phenomenal dataset to test their hypothesis. It is nice to see the lab – field – field experiment combination. I'm not sure that it answers the question they laid out, but that set up is definitely worth publishing. Calculating AS throughout these life stages will surely be an important addition to the AS field and provides an interesting alternative to the traditional approach.

We thank the Reviewer for their compliments. We understand that this was a dense manuscript, and the Reviewer's comments have been extremely helpful in ensuring that our study questions are clearly laid out in the Introduction.

Reviewer #3

General assessment

This manuscript is about the conservation biology of local Chinook salmon populations in California, USA. It explores ideas of aerobic scope limitation as a driver for fitness in nature by using a metabolic index model. The study is based on meta-analyses of previous work, and thus do not contribute any new empirical data. While it is interesting to discuss whether physiological metrics obtained in a laboratory actually is predictive of survival in wild animals, the approach used here appear overly convoluted and at times confusing owing to vaguely defined terminology and insufficiently explained theories.

We are sorry to hear the Reviewer felt our approach was confusing. We note that Reviewer 1 had a different reaction, finding the paper to be intelligible, well-written, and logical in order. Reviewer 2 also appreciated our "lab – field – field experiment combination" and how we "employed a phenomenal dataset to test [our] hypothesis." We hope that the additional changes to the text, in response to comments by Reviewers 1 and 2, improve our paper's clarity and transparency.

Considering that respirometry experiments used for obtaining aerobic scopes of fishes are incredibly sensitive to methods, equipment, protocols, animal husbandry conditions, health and nutritional status, size and life-stages, and numerous other factors, it is generally a faulty assumption that one can make useful meta-analyses of such data obtained from multiple sources and periods.

It is noteworthy that the field is rapidly changing in this regard. Over a decades' worth of literature has now conducted meta-analyses using respirometry data from multiple sources and periods (e.g., Deutsch et al. 2015, Seibel & Deutsch 2020, Penn & Deutsch 2024). Furthermore, we conducted extensive efforts to test the assumptions surrounding the pooling of multiple respirometry data sources in our modeling approach, and found no effect of respirometry methodology on our outcome of interest (lines 445-485).

Finally, we have noted that all respirometry studies used in our paper were from the same research group and therefore consistency in methodology was far greater than past meta-analyses of respirometry data (lines 431-435): "To determine the metabolic traits of juvenile

Chinook salmon, we used data from published intermittent respirometry experiments from the same research group that measured standard metabolic rate (SMR) and maximum metabolic rate (MMR) of captivity-raised fry and smolts comprising all populations (20, 34, 35, 36).”

Reference cited in this response, but not included in our manuscript:

Penn, J. L., & Deutsch, C. (2024). Geographical and taxonomic patterns in aerobic traits of marine ectotherms. *Philosophical Transactions of the Royal Society B*, 379(1896), 20220487.

The manuscript suffers from taking established and straightforward physiological concepts and making them overly complex by deriving new seemingly unnecessary parameters without really adding significant new insights. A good example is Figure 1 which makes no sense at all to me after having looked at it for quite a while. Another example is the abstract, which is vaguely written, and it is difficult to gather here what was actually done in terms of data analyses and methods.

We are confused by this comment given that all parameters we derived are well-known from a decade of metabolic index literature (e.g., Deutsch et al. 2015, Deutsch et al. 2020, Duncan et al. 2020, Howard et al. 2020, Burford et al. 2022, Essington et al. 2022, etc.). The only exception is ϕ_{stable} , the definition and relevance of which we thoroughly explain in the manuscript.

For Figure 1 – we designed this figure specifically to mimic Figure 1 in the seminal metabolic index paper (Deutsch et al. 2015), but with more labelling to help the reader. We note figures of this style have been presented in a slew of subsequent research (see references in previous paragraph). In response to a comment by Reviewer 1, we have changed “inverse kinetic energy” to “inverse temperature” and have updated all appearances of “inverse kinetic energy” in the main text and supplementary information accordingly. Furthermore, we have better explained why inverse temperature is used when parameterizing the metabolic index in Figure 1’s caption (lines 645-651): “The intercept and slope of the associations between hypoxia tolerance and inverse temperature supplied the metabolic traits A and $-E$, respectively, for each lifestage (Table 1). These traits were then used to parameterize the metabolic index (Equation 1) so that environmental temperature and O_2 could be converted into ϕ (i.e., FAS). For the units to cancel out in this equation, it is necessary for temperature to take inverse format in eV; however, for the convenience of the reader, the secondary x-axis (above) shows the corresponding temperature in °C.”

We note that Reviewer 1 reported that “the abstract was well written.” Cognizant of space limitations, we have added some detail to the Abstract. See responses to specific comments below.

Specific comments

L6: Would be useful to the reader with a short definition of the metabolic index here. It reads as if it is something else than the aerobic scope.

Please see our response to the Reviewer’s next comment.

L6 and the abstract in general: "...we related..." seems to be a rather vague description of the methodology used. Reading the abstract further it is unclear what kind of analyses were made and what factors were considered when defining a critical or stable metabolic index. It is also unclear what data was even used for these meta-analyses.

We thank the Reviewer for specific suggestions for improving the Abstract. We have repositioned "the metabolic index (ϕ)" so it is clearer that it is equivalent to AS, and added some methodological details (lines 6-12): "Using respirometry data, telemetry studies, long-term population monitoring, and in situ predator-prey experiments, we related AS to two Chinook salmon (*Oncorhynchus tshawytscha*) population bottlenecks in the wild, juvenile rearing and migration. We found that AS, which we quantified using the metabolic index (ϕ), was associated with success probability for these bottlenecks only under a relatively narrow window of viable environmental conditions, depending on intraspecific metabolic trait diversity and hydrologic conditions."

L21-22: This is a convoluted way to define aerobic scope. The aerobic scope is the difference between maximum and standard metabolic rate and is affected by environmental temperature and oxygen levels as well as numerous other factors. The definition also needs a reference. It should also be stated that this concerns aquatic ectothermic animals, mainly fish species.

We have re-written the first sentence to include that aerobic scope is aerobic activity above maintenance levels and provided references (lines 21-23): "Aerobic scope (AS) is an organism's fundamental capacity to perform aerobic work above maintenance levels given metabolic traits and environmental temperature and oxygen (O_2) supply (1, 2)."

The Reviewer provides an excellent definition for absolute aerobic scope, a specific version of aerobic scope. Our goal in introducing aerobic scope here was to provide a broader definition that covers all versions of how aerobic scope is measured (e.g., absolute and factorial). Aerobic scope the way we define it here is not specific to aquatic ectotherms, so we have not added this detail here—it comes later in the same paragraph (please see line 28).

L22-25: This sentence is way too long, and the key content described here needs to be more thoroughly fleshed out in more sentences to help the reader understand these concepts. What exactly is meant by temperature-dependent ratio of environmental oxygen supply and demand? This will also greatly depend on acclimation history, body size, health status, stress levels and many other factors.

To avoid redundancy in the manuscript, we included a more thorough definition of ϕ at the beginning of the (now) Results and Discussion (lines 127-129): " ϕ is a unitless integer directly proportional to factorial AS (FAS), or AS relative to standard metabolic rate (SMR), SMR being the energetic cost of maintenance and very limited movement in fishes (44)." Given that Reviewer 1 reported that "The introduction is overall well written," we have elected to not make any further changes to the Introduction. However, if the editor would like us to break up this sentence, we are more than happy to figure out a way to succinctly do so.

L29: This is also too vague. I assume you mean factorial aerobic scope (max metabolic rate divided by standard metabolic rate). It has been an ongoing debate whether to report absolute

or factorial aerobic scopes for some time. Both have their flaws and advantages. A range of 2-5 in factorial AS seems like such a wide range that it becomes pointless to define it as a critical threshold. Please see:

<https://doi.org/10.1007/s11160-018-9516-3>

We thank the Reviewer for drawing our attention to this reference. Indeed, we specify that factorial aerobic scope is used later in the manuscript when defining the metabolic index mathematically (please see lines 127-129). The range of critical values the Reviewer refers to here reflects that critical values depend on the species evaluated. We have added this detail (lines 27-31): “The main supporting evidence is that the warmwater limits to the current distributions of marine ectotherms are bounded by critical ϕ values (ϕ_{crit}), such that ectotherms do not inhabit marine environments unless they allow for AS at least 2-5 times greater than that required for life support (i.e., ϕ_{crit} typically ranges from 2-5 depending on the species evaluated) (3, 8).”

L82: From a species fitness point of view would it not be more relevant to focus on mature migrating adults prior to spawning? Larger and sexually mature fish are often assumed to have lower aerobic scopes. Partly owing to spending substantial resources on gonad development. We agree this would be an interesting avenue for future research, and have added this sentiment to a relevant portion of the (now) Results and Discussion (lines 377-379): “Similarly, the conservation of Chinook salmon in California’s Sacramento-San Joaquin watershed would benefit from an understanding of how AS is relevant for the success of mature adult migrations to upstream spawning grounds.”

L116-119: These concepts should be introduced properly in the introduction with references. The definition of standard metabolic rate here is inadequate. For instance, see: <https://doi.org/10.1111/jfb.12845>

We thank the Reviewer for drawing our attention to this reference and agree that this is the most relevant source by which to define SMR. This is the same reference we cite at the end of this sentence (i.e., reference 41 in our original draft [now reference 44] is the paper linked to by the Reviewer).

We have updated the definition we provide by changing “life support” to “maintenance,” which makes it more consistent with our definition of aerobic scope in the introduction (lines 127-129): “ ϕ is a unitless integer directly proportional to factorial AS (FAS), or AS relative to standard metabolic rate (SMR), SMR being the energetic cost of maintenance and very limited movement in fishes (44).”

L300: The discussion seems rather short with some less relevant paragraphs (e.g., L301-307). I particularly miss specific discussions and conclusions made from the meta analyses performed. For instance, what specific environmental conditions and aerobic scopes will actually impair these chinook salmon populations? Instead, lots of vague and confusing phrasing are used such as:

“we found that strategically expanding habitat quantity (e.g., via flow deliveries) (23, 24, 25)

would bolster the fitness of Chinook salmon even under AS-specific limitations in habitat quality.”

The discussion also does not consider potential limitations in the data analyses presented here, or what future directions would look like for linking aerobic scope with fitness in nature.

In this revision we have changed the title of the “Results” section to “Results and Discussion” and re-labeled the “Discussion” section to “Conclusions.” The Results and Discussion includes the information requested by the Reviewer, such as the environmental conditions and aerobic scopes relevant to ecological fitness.

We have also included additional paragraphs highlighting limitations to our approach and future research directions (lines 361-379). These paragraphs are the 4th and 5th in the new “Applicability of aerobic scope-based management actions in the study system” section of the (now) Results and Discussion (lines 304-379). In this new section, we succinctly discuss when and why ϕ can be a more relevant metric of water quality than temperature, connect our findings to potential management options, and highlight some limitations that could be addressed by future work.

L371-372: This is a bold statement. Various more recent studies have now shown that the chase protocol clearly underestimate the true MMR in fish species:

<https://doi.org/10.1242/jeb.246439>

And here is a study on Chinook salmon, specifically:

<https://doi.org/10.1093/conphys/coaa063>

We thank the Reviewer for directing our attention to these references. We have modified this sentence to include these more recent findings (lines 451-453): “Exhaustive chase and forced swimming protocols have been shown to yield equivalent MMR values (73) (but see 74), and we found that MMR method did not affect metabolic trait determination (Figure S4, Figure S5, Figure S6).” However, we believe our results stand because we specifically found MMR method did not affect metabolic trait determination.

L375-376: Then that would point to inadequate data. Something to consider in the discussion. Acclimation history prior to experimentations certainly will affect metabolic rates.

Ok - we have modified this sentence to include that metabolic rates can be impacted by temperature acclimation (lines 455-456): “While acclimation temperature can affect metabolic rates, we found that it did not affect metabolic trait determination (Table S5).” We appreciate the suggestion, but given that acclimation temperature did not statistically impact metabolic traits in our study, the data used are adequate and do not warrant a discussion caveat.

Dear Dr. Repetto,

Thank you for the opportunity to revise our manuscript (COMMSBIO-24-8737A) for consideration at *Communications Biology*.

In this revision, we have carefully considered the feedback provided by Reviewer #1. All author responses are colored in blue and line numbers reference the “clean” version of the revised manuscript without track changes. A version of the manuscript with track changes highlighted is also included with this revision.

As suggested by this Reviewer, we have 1) taken steps to refine the Introduction and improve conceptual framing, and 2) included more clarity in the Methods about how data were handled for the metabolic trait analysis. We have also added more explicit discussion of any relevant caveats or limitations. No Methods or Results have been changed in this revision.

Thank you again for the opportunity to revise our manuscript. We look forward to hearing from you soon.

Sincerely,
The Authors

Reviewer #1 (Remarks to the Author):

Many empirical studies identify that aerobic metabolic capacity (Aerobic scope; AS) is commonly constrained at high temperatures and low environmental oxygen levels. However, the extent to which such constraints dictate performance of fitness linked traits, habitat use, and habitat suitability in nature is challenging to assess. To help address this challenge the present study integrates several existing data sources to model the association of the predicted aerobic capacity of juvenile chinook salmon with their habitat use, survival, and predation risk by a non-native predator. The models indicate that below a certain AS further environmentally driven reductions in AS are associated with reduced habitat use and migration success, and increased probability of predation. As with other reviewers I found the study to be a highly creative, and generally impressive undertaking.

The authors provided extensive rebuttals to prior reviews, however there were a few issues raised previously or associated with those comments that in my assessment were not adequately addressed.

We thank the Reviewer for their compliments, and for their assessment of our rebuttals to prior reviews. Following their helpful suggestions, we believe we have now adequately addressed all issues raised in the prior reviews.

General comments:

At 8 paragraphs and ~1300 words I found the introduction to be unnecessarily long and that it did not follow an obvious flow. As with other reviewers, with the number of concepts the

manuscript tries to address in the introduction I found it difficult to follow or link together the core rationale. I think much of background on AS could be more direct. In particular, the logic in the paragraph from L35-46 was not obvious to me, nor is it clear specifically how this study would address this proposed dichotomy. The objective that immediately follows this paragraph does not seem to address the knowledge gap raised.

We thank the Reviewer for specifying where the Introduction could benefit from improved flow and increased clarity. In response to this comment (and the next two comments), we have made four main edits:

1. To help with flow, we have now included a few opening sentences that convey the broader conceptual relevance of AS before jumping directly into the finer details (lines 21-26).
2. To better connect our conceptual framing with the study's objectives, we have revised the 2nd and 3rd paragraphs for clarity. Please see the revised versions of these paragraphs, now the 3rd and 4th paragraphs of the Introduction (lines 40-63).
3. To help reduce the length of the Introduction, we have moved a paragraph with salmon-specific information to a new "Study System" section immediately after the Introduction (lines 111-126). This section could be moved to the Methods, but that would necessitate adding clarifying sentences to the Results and Discussion. Thus, we opted for the former option which does not involve adding length to the main text.
4. Finally, we found a few redundancies in our Introduction. By deleting these sentences, we were able to further reduce the length by several sentences.

I also think the introduction somewhat misinterprets common views among experimental biologists about the likely importance of AS in an ecological context. I can't think of any that would argue that AS governs all aspects of organismal performance (157-58). Most would agree that the extent to which AS influences fitness is context dependent in natural settings, varying with both environment and life history demands (e.g., feeding vs upriver migration) (e.g., Farrell 2016).

The Reviewer brings up a valid point. Our thinking here was that the target audience of this manuscript is not just experimental biologists. Ecologists, oceanographers, and ecosystem managers (for example) also have views on how temperature and oxygen synergistically impact where and how water-breathing taxa can exist. Thus, our interpretation of "common views" in the Introduction attempts to capture this wider swath of views while staying true the views of experimental biologists. We believe that the revised versions of the 2nd and 3rd paragraphs (now the 3rd and 4th paragraphs, lines 40-63) have clarified the conceptual framing, and thereby curbed any potential misrepresentation.

Some material specific to chinook salmon can also be greatly condensed and specifics can be raised in the methods or discussion where relevant.

This is a great suggestion – as stated previously, we have moved a paragraph with salmon-specific information to a new "Study System" section immediately after the Introduction (lines 111-126).

The other issue raised by Reviewer 3 that I thought was not adequately addressed was related to methods used in the assessments of metabolic rates. The authors make several assertions about metabolic trait assessments that are not accurate. In particular, based on its definition, 'standard metabolic rate' has specific measurement criteria that were not met, and/or not standardized in the studies from which 'SMR' was taken from. The authors of those studies are clear to state that their measurements are referred to as RMR, not SMR, because they don't meet those criteria. In general, the differences among the studies may not impact the overall conclusions but they need to be addressed. I elaborate on these below.

We appreciate the feedback from the Reviewer. Following their helpful suggestions, we have made edits to the Methods to increase the clarity and accuracy of how we describe and qualify our metabolic trait determination (see our responses to the relevant comments below). No Methods or Results have been changed.

Specific comments:

1)L88-92: This sentence could be shortened/split for clarity.

Done, thanks (lines 113-117): "The subsequent transition to the migratory smolt lifestage, which must traverse low-elevation freshwater habitats such as the Delta, is characterized by biochemical changes involved in osmoregulation and metabolism (23, 41). These changes prepare individuals for a pelagic lifestyle in the California Current, an Eastern Boundary Upwelling Ecosystem characterized by relatively shallow layers of cold, hypoxic water (5, 8)."

2)L98-101: simplify use of 'returning' to improve readability

Done, thanks (lines 123-126): "During fall and winter, fry of winter-, spring-, and fall-returning adults are usually present, as well as smolts of winter-returning adults. During spring and summer, fry of late fall-returning adults and smolts of spring-, fall-, and late fall-returning adults tend to be present (30, 44)."

3)L431: This information is intended to rebut a comment from Reviewer 3 regarding difficulties in comparing absolute values across studies. However, despite being from the group, the cited studies use quite different methods including different types of respirometers (swimming vs. static), measurement cycle durations, fasting durations, and measurements were performed over several years. In the end, the current investigation may be robust to these differences, but it cannot be implied that these studies used highly comparable methods. These methods are also not standard in the field as implied by referring to them as "established".

Following the Reviewer's comment, we have added an overview sentence here clarifying the differences in methodologies used in these studies. We also note that the manuscript includes more details on how the cited studies differ in their methodology (lines 440-442): "These studies were performed over several years and had slight methodological differences (e.g., respirometers, measurement cycle durations, and fasting durations), which are discussed in more detail below."

4)L 435: SMR, MMR, and Aerobic scope can be repeatable in some contexts, but it is not a rule as stated here.

Norin, T., & Malte, H. (2011). Repeatability of standard metabolic rate, active metabolic rate and aerobic scope in young brown trout during a period of moderate food availability. *Journal of Experimental Biology*, 214(10), 1668-1675.)

OK. We have changed “are” to “can be” in this sentence (lines 442-444): “SMR, MMR, and resultant aerobic scope (AS) can be repeatable traits (68) that are posited to allow for a more nuanced assessment of how species interact with their environment (69).”

5)L441: Fasting durations were as short as 24 hours

Great catch, thanks. The original values reported in our manuscript included the additional fasting that occurred during acclimation. To avoid confusion, we have updated the range as reported in the text of these sources as suggested by the Reviewer (lines 447-449): “Following (45), SMR values were calculated using metabolic rates gathered on fasted fish (24-48 hours of fasting depending on temperature) during an overnight period (20, 31, 32, 33).”

6)L442: Mo2 measurement periods for RMR or SMR varied between 12 and 24 hours across studies to the point that the authors go out of their way to highlight that they use the term ‘RMR’ in 3 out of 4 of these studies rather than SMR because their approach was not suitable to reliably/confidently estimate SMR. Inflations of SMR may be particularly consequential when being used to calculate FAS

Halsey, L. G., Killen, S. S., Clark, T. D., & Norin, T. (2018). Exploring key issues of aerobic scope interpretation in ectotherms: absolute versus factorial. *Reviews in Fish Biology and Fisheries*, 28(2), 405-415.

We thank the Reviewer for pointing this out, and have clarified the measurement durations in the Methods. We have also clarified the potential for SMR values to be inflated (with a resulting impact on FAS), but expanded upon our interpretation that any variation in SMR attributable to a short measurement duration is unlikely to impact our overall results (lines 449-455): “SMR was determined over measurement periods that ranged from 12-24 hours in these studies, which is considered relatively short (45). Any inflation of SMR attributable to short measurement periods could be influential when determining factorial AS (FAS) (70). However, fish were monitored during all respirometry trials using infrared cameras; outside of small fin movements to maintain position in respirometers (45), fish movement was excluded from SMR values. Thus, the values used were considered a decent approximation of SMR and therefore unlikely to impact FAS, nor the overall results of our study.”

7)L 451: These swimming protocols differ from those tested in the cited validation studies and there are other validation studies that show the similarity of these assessments varies greatly with analytical methods:

Little, A. G., Dressler, T., Kraskura, K., Hardison, E., Hendriks, B., Prystay, T., ... & Eliason, E. J. (2020). Maxed out: optimizing accuracy, precision, and power for field measures of maximum metabolic rate in fishes. *Physiological and Biochemical Zoology*, 93(3), 243-254.

Zhang, Y., Gilbert, M. J., & Farrell, A. P. (2020). Measuring maximum oxygen uptake with an incremental swimming test and by chasing rainbow trout to exhaustion inside a respirometry chamber yields the same results. *Journal of Fish Biology*, 97(1), 28-38.

We thank the Reviewer for providing these references, which show that different protocols can produce similar estimates of MMR, which in our opinion supports our methodology. Given that our data sources used the same protocol, swim tunnels, data sheets, and base code as the cited validation studies, we hope that the Reviewer will agree. We have taken this opportunity to add some methodological details to the text regarding the forced swimming protocol (lines 460-462): “During forced swimming, which followed a modified U_{crit} protocol (73, 74), individuals were required to swim against a stepwise increasing current in a swim tunnel respirometer until exhaustion.”

8)L 463: These studies allowed [O₂] to drop as low as 80% air saturation and in these types of studies [O₂] rarely reaches 100% air saturation during flushing. The statement that ‘SMR and MMR were measured at 21kPa’ is thus not accurate. I would assume whatever corrective action or sensitivity analysis would be done to address this would show a similar pattern of results, but it should be addressed regardless.

We appreciate the Reviewer drawing our attention to this, and have updated the text to reflect that SMR and MMR were measured as close to 21 kPa as possible (lines 472-475): “SMR and MMR were measured at as close to 21kPa (air saturation of O₂ at sea level) as possible in all intermittent respirometry experiments (20, 31, 32, 33), which is standard practice for salmonids and other normoxic species (13). We therefore assumed that O_{2crit} of MMR was equal to 21kPa and then solved for O_{2crit} of SMR.”

9)L466: This is as not as universally accepted as stated here. I would state that “we assumed O_{2crit} of MMR to be 21kPa...”

Done, thanks (see previous response).

Reviewer #2 (Remarks to the Author):

This study cleverly links aerobic scope with ecological fitness in wild Chinook salmon in California by calculating how metabolic index relates to fry rearing and smolt migration. The research integrated laboratory experiments, fine-scale field tracking, and predator studies to link physiology with management-relevant outcomes.

I commend the authors for the creativity they employed in this study. I think it’s great for the field of ecophysiology that we’re seeing more studies embracing meta-analysis and data synthesis of physiological data. In general, I quite enjoyed reading the manuscript.

I was asked specifically to gauge the author’s rebuttal to reviewer 1’s comments. Overall, I found Reviewer 1’s criticisms valid and constructive. I also found the author’s responses corresponding edits appropriate and satisfactory. I could not find an instance where I found the author’s response unsatisfactory. I would suggest that the revised manuscript is appropriate for publication in its current form.

We thank the Reviewer for their compliments, and for their assessment of our rebuttals to prior reviews. We are very glad to hear they enjoyed reading the manuscript!